# A ribose-functionalized NAD$^+$ with unexpected high activity and selectivity for protein poly-ADP-ribosylation

Xiao-Nan Zhang[1,7], Qinqin Cheng[1,7], Jingwen Chen[1], Albert T. Lam[1], Yanran Lu[1], Zhefu Dai[1], Hua Pei[2], Nikolai M. Evdokimov[3], Stan G. Louie[2] & Yong Zhang [1,4,5,6]

Nicotinamide adenine dinucleotide (NAD$^+$)-dependent ADP-ribosylation plays important roles in physiology and pathophysiology. It has been challenging to study this key type of enzymatic post-translational modification in particular for protein poly-ADP-ribosylation (PARylation). Here we explore chemical and chemoenzymatic synthesis of NAD$^+$ analogues with ribose functionalized by terminal alkyne and azido groups. Our results demonstrate that azido substitution at 3'-OH of nicotinamide riboside enables enzymatic synthesis of an NAD$^+$ analogue with high efficiency and yields. Notably, the generated 3'-azido NAD$^+$ exhibits unexpected high activity and specificity for protein PARylation catalyzed by human poly-ADP-ribose polymerase 1 (PARP1) and PARP2. And its derived poly-ADP-ribose polymers show increased resistance to human poly(ADP-ribose) glycohydrolase-mediated degradation. These unique properties lead to enhanced labeling of protein PARylation by 3'-azido NAD$^+$ in the cellular contexts and facilitate direct visualization and labeling of mitochondrial protein PARylation. The 3'-azido NAD$^+$ provides an important tool for studying cellular PARylation.

[1] Department of Pharmacology and Pharmaceutical Sciences, School of Pharmacy, University of Southern California, Los Angeles, CA 90089, USA. [2] Titus Family Department of Clinical Pharmacy, School of Pharmacy, University of Southern California, Los Angeles, CA 90089, USA. [3] Department of Chemistry and Biochemistry, University of California, Los Angeles, CA 90095, USA. [4] Department of Chemistry, Dornsife College of Letters, Arts and Sciences, University of Southern California, Los Angeles, CA 90089, USA. [5] Norris Comprehensive Cancer Center, University of Southern California, Los Angeles, CA 90089, USA. [6] Research Center for Liver Diseases, University of Southern California, Los Angeles, CA 90089, USA. [7] These authors contributed equally: Xiao-Nan Zhang, Qinqin Cheng. Correspondence and requests for materials should be addressed to Y.Z. (email: yongz@usc.edu)

Numerous biological processes are orchestrated by protein post-translational modifications (PTMs)[1]. Among key PTMs is protein ADP-ribosylation catalyzed by a superfamily of enzymes named ADP-ribosyltransferases (ARTs) with nicotinamide adenine dinucleotide (NAD⁺) as a cosubstrate. The human genome is found to encode 20 ART enzymes including intracellular poly-ADP-ribose polymerases (PARPs) and sirtuins (SIRTs) and ecto-ART1-5[2], which possess poly- or mono-ADP-ribosylation activity. Protein ADP-ribosylation plays vital roles in regulating genome stability, protein homeostasis, cell proliferation, differentiation and apoptosis[2–5]. In the human ART superfamily, 17 members (85%) are strongly implicated in various human diseases[5–19]. The activation or enhancement of those ART enzymatic activities positively correlate with pathogenesis and progression of those diseases. Given their importance and emerging roles in a wide range of prevalent human diseases, ART enzymes have attracted considerable interest for early diagnosis and treatment of disease. To study NAD⁺-dependent ADP-ribosylation, a number of NAD⁺ analogues were chemically synthesized, which are featured with modified adenines by alkyne, biotin, and fluorescent groups[20–27] to label and profile substrate proteins and/or visualize cellular ADP-ribosylation. The established NAD⁺ analogues with substituted adenines raise the question of whether functional groups can be attached to other positions for generating NAD⁺ molecules with enhanced and/or new functions.

Herein, we explore the generation of a series of ribose-functionalized NAD⁺ analogues using chemical and chemoenzymatic approaches (Fig. 1). It is found that recombinant human nicotinamide riboside kinase 1 (NRK1) and nicotinamide mononucleotide adenylyltransferase 1 (NMNAT1) can efficiently catalyze formation of 3′-azido NAD⁺ from its nicotinamide riboside (NR) analogue precursor in the presence of adenosine triphosphate (ATP). Enzyme- and cell-based assays indicate that 3′-azido NAD⁺ displays excellent substrate activity for protein poly-ADP-ribosylation (PARylation), comparable to that of native NAD⁺. Importantly, in comparison to established adenine-modified NAD⁺ analogues with strong activities for ADP-ribosylation, the 3′-azido NAD⁺ displays unexpected higher activity and selectivity for protein PARylation catalyzed by human PARP1 and PARP2. Moreover, the 3′-azido NAD⁺-based poly-ADP-ribose (PAR) polymers reveal improved resistance to degradation by human poly(ADP-ribose) glycohydrolase (PARG). These unique properties for the 3′-azido NAD⁺ collectively result in more significant labeling of protein PARylation in the cell lysates relative to NAD⁺ and adenine-substituted NAD⁺ analogues and enable direct visualization and labeling of mitochondrial protein PARylation. This ribose-functionalized NAD⁺ with marked activity and specificity for protein PARylation not only offers an important tool for studying post-translational ADP-ribosylation but also may pave the ways

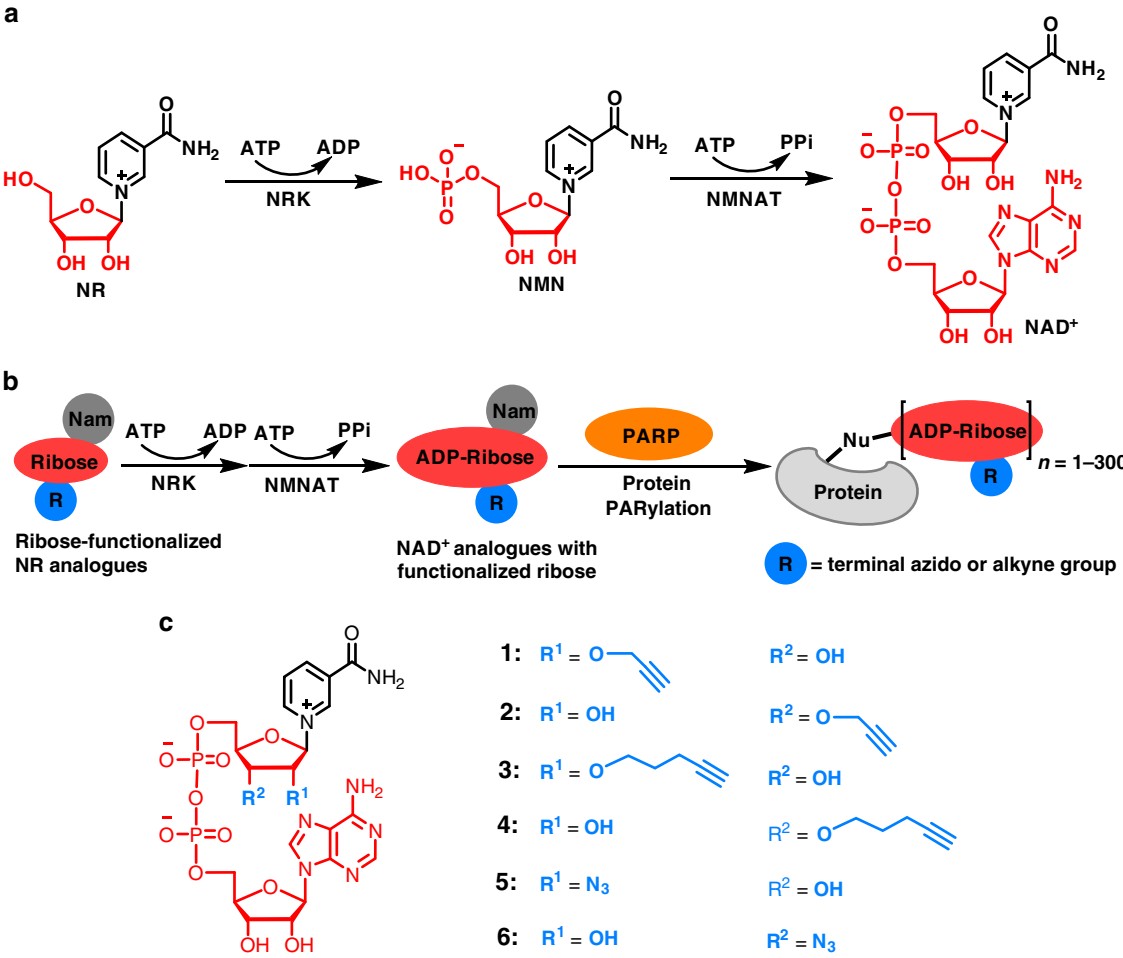

**Fig. 1** Biosynthesis of NAD⁺ and ribose-functionalized NAD⁺ analogues. **a** NRK- and NMNAT-catalyzed formation of NAD⁺. **b** Enzymatic synthesis of NAD⁺ analogues with functionalized riboses for protein PARylation. **c** Chemical structures of NAD⁺ analogue **1–6**. PPi: pyrophosphate; Nam: nicotinamide; Nu: nucleophile

toward the creation of bi- and multi-functional $NAD^+$ molecules for investigating PARylation-dependent signaling and processes.

## Results

### Chemical and enzymatic synthesis of $NAD^+$ analogues.

Inspired by NRK- and NMNAT-mediated biosynthesis of $NAD^+$ from NR (Fig. 1a), we envisioned that ribose-functionalized NR may allow facile chemoenzymatic synthesis of $NAD^+$ analogues (Fig. 1b). Human NRK1 and NMNAT1 display adequate catalytic activities for NR + ATP and nicotinamide mononucleotide (NMN) + ATP, respectively, and promiscuity toward other substrate analogues[28–30]. X-ray structures of human NRK1 and NMNAT1 support that ribose-functionalized NR and NMN are likely recognized for catalysis[31,32] (Supplementary Fig. 1). Moreover, previous studies showed that leaving group activation is an important catalytic force for reactions catalyzed by multiple $N$-ribosyltransferases[33–38]. Combined with structural analysis of PARP active sites, these results suggest that modifications of NR ribosyl 2′-OH and 3′-OH with terminal alkyne and azido groups can possibly result in bioorthogonal $NAD^+$ with considerable substrate activities and a facile approach for the production of $NAD^+$-based chemical tools. Importantly, such ribose-functionalized $NAD^+$ may allow incorporation of additional functional groups at distinct positions without signicant loss of substrate activity for the development of $NAD^+$ molecules with dual or multiple functions.

To test this notion, propargyl, pentynyl, and azido were chosen for modifications of NR 2′-OH and 3′-OH to generate **1–6** (Fig. 1c and Supplementary Figs. 2–8). The stereochemistries of the generated intermediates O-benzoyl protected NR1-6 were determined as β-isomers on the basis of $^1H$–$^1H$ COSY experiments to confirm proton assignments and subsequent NOESY experiments (Supplementary Figs. 9–21). The resulting NR1-6 were then subjected to 5′-phosphorylation and pyrophosphate coupling with adenosine monophosphate (AMP) for chemical synthesis of **1–6** (Supplementary Figs. 2–8).

To explore enzymatic synthesis of **1–6** from their corresponding NR and NMN analogues, human NRK1 and NMNAT1 were expressed and purified from *Escherichia coli* (Supplementary Table 1 and Supplementary Fig. 22). In vitro biosynthesis of $NAD^+$ from NR was first carried out using purified NRK1 and NMNAT1. In the presence of NRK1 and NMNAT1 and ATP, a substantial amount of $NAD^+$ was formed from NR after 40 h incubation (Fig. 2a, b). Then, enzymatic syntheses of **1–6** were attempted under the same conditions. It was found that a significant amount of **5** and **6** could be generated by incubating NR5 and NR6 with ATP and NRK1 and NMNAT1 at room temperature for 24 or 40 h (Supplementary Fig. 23 and Fig. 2c, d), while incubation of other NR analogues NR1-4 with ATP and the purified enzymes gave no formation of $NAD^+$ analogues **1–4** (Supplementary Fig. 23). Compared with an 83% isolated yield for biosynthesis of $NAD^+$ from NR (Supplementary Fig. 24 and Fig. 2a, b), the two-step enzymatic approach gave rise to a 68% isolated yield for the production of **6** starting from NR6 (Supplementary Fig. 25 and Fig. 2c, d). Using this enzymatic method, 12.2 mg of **6** was facilely produced and puried for the later experiments. In contrast, chemical synthesis of **6** from NR6 revealed a combined yield of 32% (Supplementary Fig. 8) and the pyrophosphate coupling step needs four days plus tedious and challenging HPLC purification. These results demonstrate a facile and efficient chemoenzymatic approach for generating 3′-azido $NAD^+$.

Additionally, NR1-6 and NMN1-6 were examined separately with purified NRK1 and NMNAT1 to determine their substrate activities for enzymatic conversions. Compared with NRK1 that

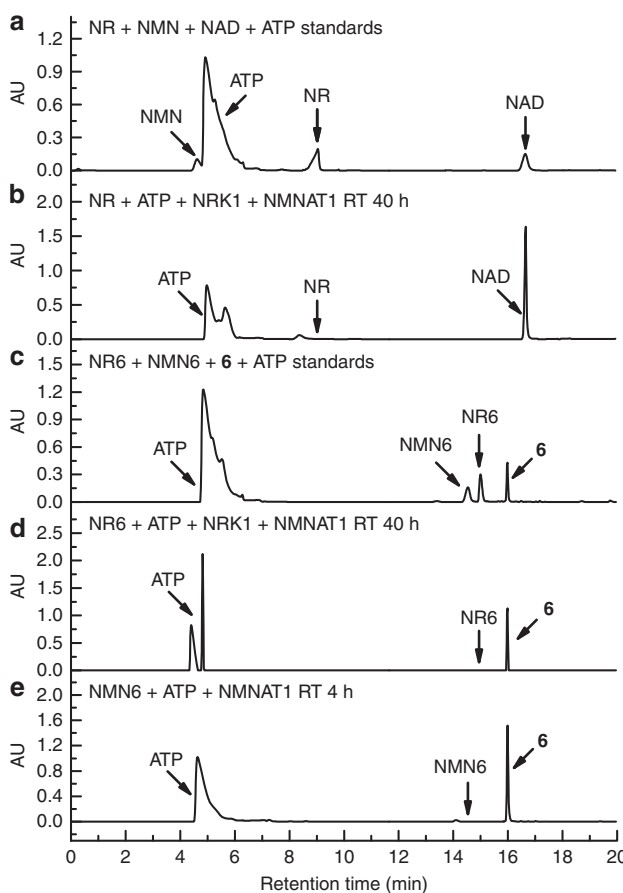

**Fig. 2** HPLC analysis of enzymatic formation of $NAD^+$ and **6**. **a** Assigned peaks for standard compounds of NR, NMN, $NAD^+$ and ATP. **b** One millimolar NR was incubated with 5 mM ATP, 5 μM NRK1, and 5 μM NMNAT1 at RT for 40 h, followed by HPLC analysis. **c** Assigned peaks for standard compounds of NR6, NMN6, **6** and ATP. **d** One millimolar NR6 was incubated with 5 mM ATP, 5 μM NRK1, and 5 μM NMNAT1 at RT for 40 h, followed by HPLC analysis. **e** One millimolar NMN6 was incubated with 5 mM ATP, 5 μM NRK1, and 5 μM NMNAT1 at RT for 4 h, followed by HPLC analysis. UV absorbance was measured at 260 nm. AU: absorbance unit

could only catalyze conversion of NR5 and NR6, NMNAT1 displayed higher tolerance to these ribosyl modifications and was shown to catalyzes formation of **1**, **2**, **5** and **6** from respective NMN precursors (Supplementary Figs. 26–28). HPLC analysis revealed that NMN6 could be rapidly converted to **6** within 4 h in a 74% yield at the milligram level (Supplementary Fig. 29 and Fig. 2e). These results indicate that the azido substitution at NR 3′-OH position allows efficient enzymatic synthesis of **6**, in particular from its NMN analogue precursor.

### Substrate activities of $NAD^+$ analogues for human PARP1.

To evaluate substrate activities of **1–6** for protein PARylation, full-length human PARP1 was expressed and purified from *E. coli*. Human PARP1 is a well-characterized enzyme for protein PARylation. In addition to catalyzing PARylation on various types of cellular proteins, PARP1 also undergoes robust auto-PARylation in the presence of $NAD^+$ and DNA fragments. The substrate activities of **1–6** were first evaluated by performing PARP1-catalyzed automodification with highly pure **1–6** (Supplementary Fig. 30 and Supplementary Table 2). It was shown that both $NAD^+$ and **6** resulted in strong auto-PARylation of PARP1, which could be potently suppressed by veliparib, an inhibitor of PARP1-PARP4 (Fig. 3a). This also indicates that the

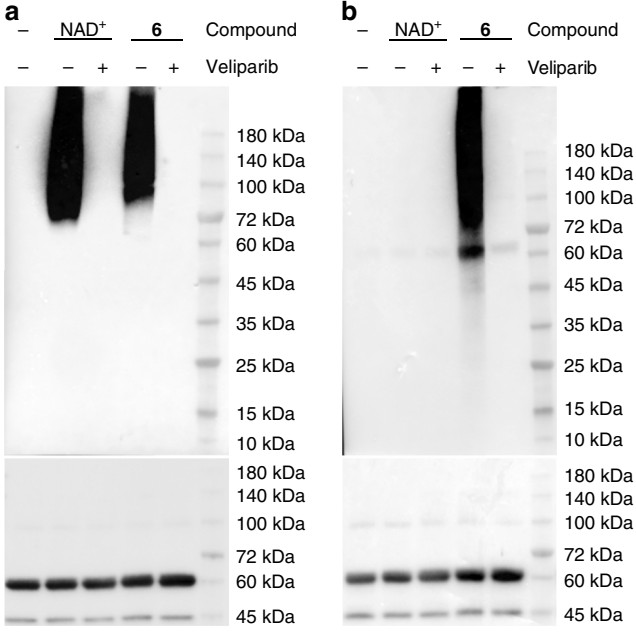

**Table 1 Kinetic parameters of NAD⁺ and its analogues for human PARP1**

|  | Substrate | $k_{cat}$ (min⁻¹)[a] | $K_m$ (μM)[a] | $k_{cat}/K_m$ (min⁻¹ M⁻¹) |
|---|---|---|---|---|
| PARP activity | NAD⁺ | 4.7 ± 0.4 | 145.4 ± 36.0 | 3.2 × 10⁴ |
|  | 2 | 0.06 ± 0.01 | 218.1 ± 81.7 | 2.8 × 10² |
|  | 6 | 4.1 ± 0.6 | 370.5 ± 104.8 | 1.1 × 10⁴ |
|  | 6-a-NAD⁺ | 1.9 ± 0.5 | 603.1 ± 307.9 | 3.2 × 10³ |
|  | 2-a-NAD⁺ | 1.4 ± 0.4 | 400.8 ± 238.2 | 3.5 × 10³ |
| NADase activity | NAD⁺ | 0.39 ± 0.08 | 471.5 ± 187.4 | 8.3 × 10² |
|  | 2 | 0.02 ± 0.01 | 661.1 ± 489.0 | 0.3 × 10² |
|  | 6 | 0.46 ± 0.17 | 326.9 ± 278.8 | 1.4 × 10³ |
|  | 6-a-NAD⁺ | 0.36 ± 0.02 | 370.8 ± 47.5 | 9.7 × 10² |
|  | 2-a-NAD⁺ | 0.47 ± 0.13 | 545.9 ± 313.5 | 8.6 × 10² |

[a]Data are shown as mean ± standard deviation of three replicates

**Fig. 3** Substrate activities of NAD⁺ and **6** for human PARP1. Auto-PARylation of PARP1 with 400 μM NAD⁺ or **6** was analyzed by immunoblots using an anti-PAR antibody (**a**) and a streptavidin-HRP conjugate following biotin conjugation via click chemistry (**b**). PARP1 loading controls were detected by an anti-His₆ antibody. Source data are provided as a Source Data file

anti-PAR antibody can recognize the 3′-azido NAD⁺-derived PAR polymers. Since **6** contains an azido group, which is bioorthogonal to biotin-conjugated alkynes through click chemistry, auto-modified PARP1 by NAD⁺ and **6** were then clicked with biotin and detected by a streptavidin-HRP conjugate. Consistent with the immunoblot results using the anti-PAR antibody, PARP1 modified by **6** revealed strong PARylation signals that were sensitive to veliparib treatment (Fig. 3b). Due to the lack of azido group for NAD⁺, no PARylation signals were observed for NAD⁺-modified PARP1 on this biotinylation-based immunoblot. In comparison, **5** with an azido group attached to NR 2′-OH displayed no PARylation signals (Supplementary Figs. 31 and 32). Similarly, **1** and **3** containing 2′-substitutions were shown to have no substrate activities for PARP1-catalyzed PARylation. In contrast, **2** and **4** with terminal alkyne groups modified at NR 3′-OH display strong to moderate activities for PARP1 automodification, respectively (Supplementary Figs. 31 and 32).

Next, kinetic parameters of NAD⁺, **2**, and **6** for PARP1-catalyzed automodification (PARP activity) and hydrolysis (NADase activity) were determined by HPLC-based activity assays (Table 1). The $k_{cat}$ of **6** for PARP1 auto-PARylation is 4.1 ± 0.6 min⁻¹ (mean ± standard deviation of three replicates), only slightly lower than that of NAD⁺ (4.7 ± 0.4 min⁻¹). The $K_m$ (370.5 ± 104.8 μM) of **6** is higher than that (145.4 ± 36 μM) of NAD⁺. The $k_{cat}$ of **2** is significantly lower than those of NAD⁺ and **6**. Same as NAD⁺, **2** and **6** could undergo slow hydrolysis catalyzed by PARP1. Similar to their PARP activities, the NADase activity for **6** is comparable to that of NAD⁺. These results are consistent with immunoblot analyses and support 3′-azido NAD⁺ as an excellent substrate for PARP1-catalyzed PARylation.

**Biological activities of 2 and 6 in cellular context.** To evaluate their biological activities in the cellular context, **2** and **6** were introduced into HeLa cells through transient permeabilization of

the cell membrane, which has no significant effect on cell viability[27,39,40] (Supplementary Fig. 33). Cells without transient permeabilization were used as controls (Supplementary Figs. 34 and 35). Confocal microscopic analysis of cells treated with **6** clearly revealed the extent and location of cellular ADP-ribosylation (Fig. 4a). Hydrogen peroxide (H₂O₂) can induce oxidative DNA damage in the nucleus, leading to activated PARP1 and PARP2 for DNA repair. Consistent with previous results, treatment of cells with H₂O₂ resulted in significantly increased ADP-ribosylation in the nucleus, which were suppressed by veliparib. In addition, immunoblot analyses of the lysates of HeLa cells showed marked protein labeling for cells treated with **2** and **6** (Fig. 4b and Supplementary Fig. 36). Consistent with the confocal imaging study, treatment with H₂O₂ caused significantly increased protein labeling, possibly due to the activated PARP1 and PARP2. Additions of veliparib inhibited protein labeling detected in the absence and presence of H₂O₂. Compared with imaging and labeling signals derived from **2** (Supplementary Fig. 36), the analogue **6** exhibited improved signal-over-background and sensitivity for both imaging and protein labeling studies, consistent with its excellent substrate activity for PARP1 as determined from immunoblot and enzyme kinetic assays. Collectively, these results support 3′-azido NAD⁺ as a valuable tool for studying cellular PARylation.

**Comparison of 6 and 2 with adenine-modified NAD⁺ analogues.** The substrate activity of **6** and **2** were then compared with previously reported NAD⁺ analogues with modified adenines (Fig. 5). For this purpose, 6-alkyne-NAD⁺ (6-a-NAD⁺) and 2-alkyne-NAD⁺ (2-a-NAD⁺) were synthesized according to published methods (Fig. 5b)[41,42], which show strong activities for PARP1[24,27,41,42]. Relative to **6**, 6-a-NAD⁺ and 2-a-NAD⁺ exhibit comparable $K_m$, but significantly lower $k_{cat}$ for PARP1 auto-PARylation (Table 1). The immunoblot analyses of their substrate activities for PARP1-catalyzed auto-PARylation showed consistent results that PARP1 modified by **6** displayed significantly higher PARylation signals (Fig. 5c). In addition to PARP1, these NAD⁺ analogues were examined for substrate activities for its close relative PARP2 that catalyzes protein PARylation. Similar to PARP1, human PARP2 auto-PARylated by **6** showed significantly higher signals relative to 6-a-NAD⁺ and 2-a-NAD⁺ (Fig. 5d). Furthermore, the substrate activities of **2**, **6**, 6-a-NAD⁺, and 2-a-NAD⁺ for catalytic domains of human PARP5a and PARP10 were compared (Fig. 5e, f), which catalyze protein PARylation and mono-ADP-ribosylation (MARylation), respectively. Unlike 6-a-NAD⁺ and 2-a-NAD⁺ that displayed strong automodification activities for both enzymes, **2** and **6** revealed little activities

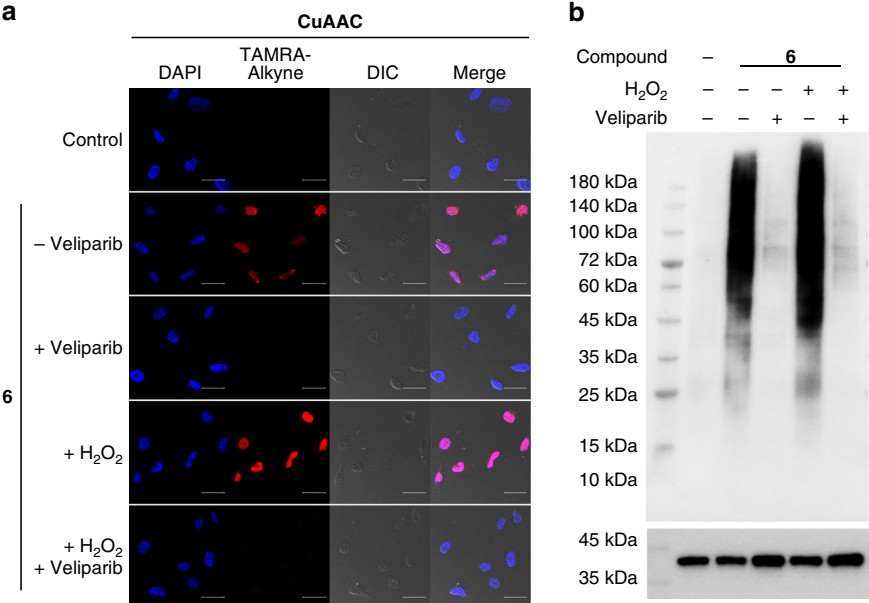

**Fig. 4** Visualization and labeling of cellular protein PARylation by **6**. HeLa cells were pretreated for 20 min in the absence and presence of $H_2O_2$ and 1 μM veliparib. Following transient permeabilization with 0.01% Triton-X-100, cells were incubated with 100 μM **6** for 45 min, followed by **a** confocal microscopic analysis through fixation, permeabilization, and fluorescent staining via click chemistry, CuAAC: copper(I)-catalyzed azide alkyne cycloaddition. DIC: differential interference contrast. Scale bars: 20 μm, and by **b** immunoblot analysis of cell lysates as detected by a streptavidin-HRP conjugate following biotin conjugation via click chemistry. GAPDH as protein loading controls were detected by an anti-GAPDH antibody. Source data are provided as a Source Data file

for auto-ADP-ribosylation catalyzed by PARP5a and PARP10 under the same conditions. These results reveal that **6** is a more selective substrate with excellent activity for PARP1 and PARP2.

In addition to their substrate activities, the stability of the PAR polymers formed by these $NAD^+$ analogues were compared upon treatment by human PARG, one of the major enzymes responsible for degrading PAR polymers (Fig. 6). In contrast to $NAD^+$- and 6-a-$NAD^+$-derived PAR polymers that were rapidly removed by PARG, 2-a-$NAD^+$- and **6**-based PAR polymers show comparable resistance to PARG-mediated degradation.

Next, **6** was applied to label protein PARylation in cell lysates for comparion with those labeled by $NAD^+$ and 2-a-$NAD^+$, which causes higher level of labelling than 6-a-$NAD^+$ (Fig. 7a, b). By using a WWE domain-based reagent[48] that exhibits comparable binding toward PAR polymers formed by $NAD^+$, **6**, and 2-a-$NAD^+$ (Supplementary Fig. 37), immunoblot analyses indicated that **6** results in more significant labelling of protein PARylation in cell lysates relative to $NAD^+$ and 2-a-$NAD^+$ (Fig. 7c, d).

**Analysis of mitochondrial protein PARylation by** 6. To further demonstrate its utility, **6** was applied to examine mitochondrial protein PARylation. In addition to the nucleus as the predominant subcellular location for protein PARylation, multiple studies indicated the presence of PARylation in mitochondria[49–54]. In vitro biochemical assays also confirmed several mitocohondrial proteins as PARP1 substrates[24]. Despite emerging but debating roles of PARylation in regulating mitochondrial DNA metabolism, limited approaches are available for studying mitochondrial PARylation and activity-based probes have yet to be developed for analyzing mitochondrial PARylation. Considering its high activity and specificity for protein PARylation, **6** was attempted to label mitochondrial PARylation in cells permeabilized with 0.025% Triton X-100. Confocoal microscopy indicated that in addition to the predominant PARylation in nucleus, considerable PARylation signals were colocalized with mitofilin, a

mitochondrial inner membrane protein (Fig. 8a), suggesting the presence of PARylation in mitochondria. Consistent with confocal imaging results, immunoblot analysis of the isolated mitochondria fractions clearly revealed significant protein labeling in the presence of **6**. And treatment of cells with $H_2O_2$ resulted in increased protein labeling by **6** in the mitochondria fractions, which were suppressed by veliparib inhibitor (Fig. 8b). Moreover, a considerable amount of PARP1 was detected in mitochondrial fractions (Fig. 8b), which is consistent with previous studies[52–54] and suggests its important role in catalyzing protein PARylation in mitochondria. Taken together, these results provide direct evidence for mitochondrial PARylation through an activity-based probe, demonstrating **6** as a valuable tool for studying cellular PARylation.

## Discussion

In summary, a 3′-azido $NAD^+$ with unexpected high activity and selectivity for protein PARylation was generated. By exploiting recombinant human NRK1 and NMNAT1, this ribose-functionalized $NAD^+$ molecule could be enzymatically synthesized in high efficiency and yields from its NR and NMN analogue precursors, establishing a facile approach for the production of 3′-azido $NAD^+$. In contrast to established $NAD^+$ analogues, **6** may represent a robust tool with enhanced activity and sensitivity for studying cellular PARylation. The enhanced labelling by **6** likely results from faster kinetics for PARylation and increased resistance to PAR removal by PARG.

The lack of substrate activity for **5** could be possibly caused by the blocked formation of branched PAR, which requires access to free 2′-OH of NR for covalently attaching ADP-ribose units, and/or the significantly decreased cleavage rate of the N-glycosidic bond resulting from adjacent electron-withdrawing 2′-azido group[55]. Substitutions of NR 3′-OH for terminal alkyne and azido groups result in $NAD^+$ analogues recognized by PARP1 for PARylation. The extended loop (G876-I895) at PARP1 active site may accommodate 3′-OH modifications (Fig. 5a) and the

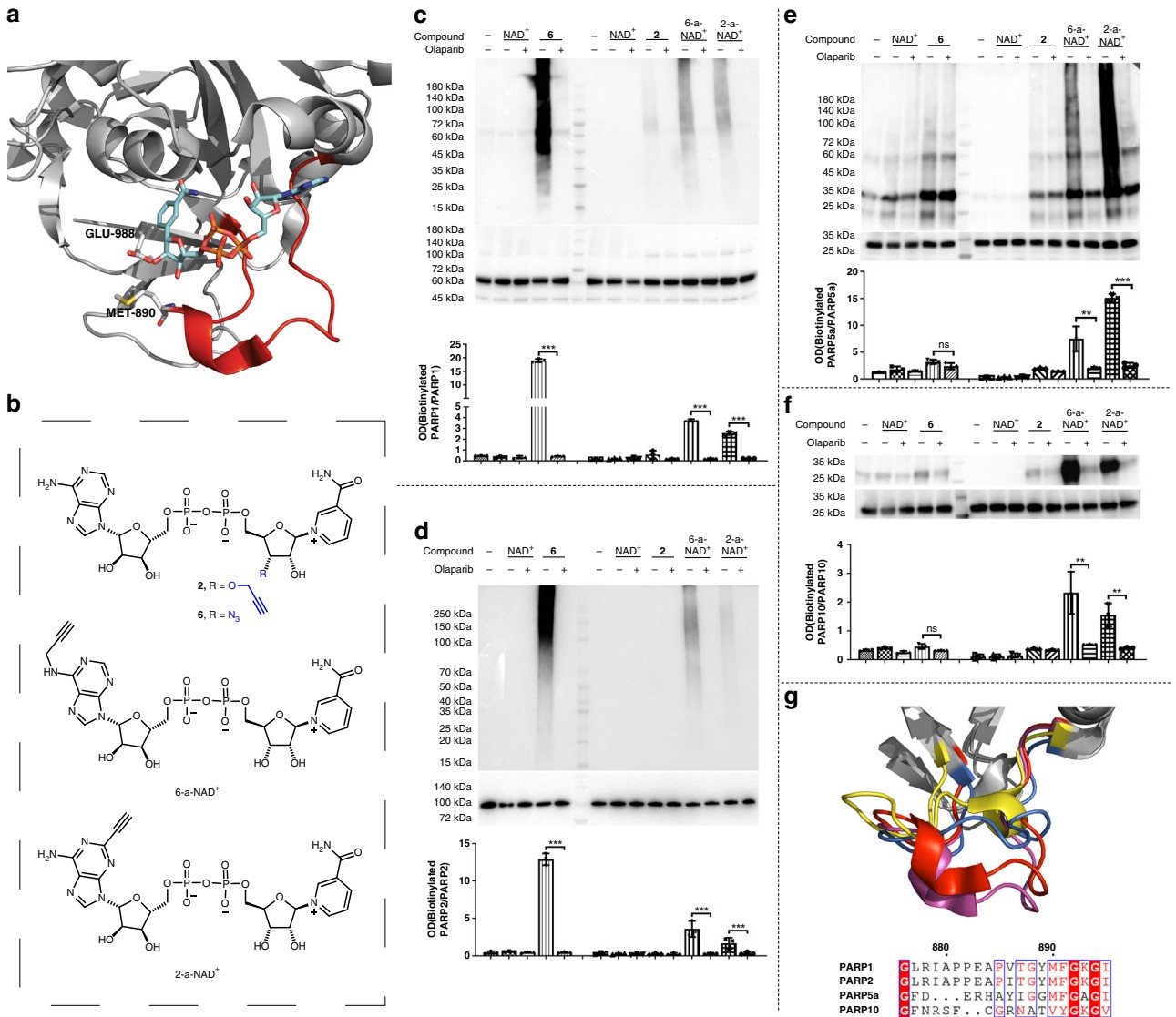

**Fig. 5** Substrate activities of NAD$^+$ analogues for human PARPs. **a** X-ray structure of human PARP1 catalytic domain in complex with benzamide adenine dinucleotide (BAD) (PDB ID: 6BHV)[43]. Residues potentially interacting with 2′- and 3′-OH groups of the benzamide riboside are shown as sticks. The loop (G876-I895) at the active site of human PARP1 is shown in red. **b** Chemical structures of **2**, **6**, 6-a-NAD$^+$, and 2-a-NAD$^+$. **c-f** Substrate activities of **2**, **6**, 6-a-NAD$^+$, and 2-a-NAD$^+$ for **c** human PARP1; **d** human PARP2; **e** catalytic domain of human PARP5a; and **f** catalytic domain of human PARP10. Automodifications of PARP1, PARP2, PARP5a and PARP10 with NAD$^+$ analogues were carried out by incubating PARP enzymes with NAD$^+$ analogues in the absence and presence of 100 μM olaparib at 30 °C for 2 h. The reaction mixtures were then labeled with azido-biotin (for NAD$^+$, **2**, 6-a-NAD$^+$, and 2-a-NAD$^+$) or alkyne-biotin (for NAD$^+$ and **6**) through click chemistry, followed by immunoblot analysis as detected by the streptavidin-HRP conjugate. PARP1, PARP2, PARP5a or PARP10 catalytic domain loading controls were detected by an anti-His$_6$ antibody or anti-PARP2 antibody. Automodification of PARP enzymes normalized to respective PARPs was analyzed by densitometry. ns: not significant, $P > 0.05$; **$P < 0.01$; and ***$P < 0.001$ by one-tailed unpaired $t$-test. Error bars represent standard deviation of three replicates. **g** The loops at the catalytic sites of PARP1, PARP2, PARP5a and PARP10. Loops were overlaid based on the solved X-ray structures of PARP1 (red; PDB ID: 6BHV)[43], PARP2 (magenta, PDB ID: 4ZZX)[44], PARP5a (yellow; PDB ID: 3UH4)[45] and PARP10 (blue; PDB ID: 5LX6)[46]. Primary sequences were aligned for the loops at the catalytic sites of PARP1 (G876-I895), PARP2 (G442-I461), PARP5a (G1196-I1212) and PARP10 (G901-V918)[47]. Source data are provided as a Source Data file

increased size and/or flexibility of the functional groups could cause decreased substrate activities for PARP1.

The high specificity and activity of **6** for PARP1 and PARP2 could be attributed to the extended loop (G876-I895) and (G442-I461) at PARP1 and PARP2 active sites, respectively, which possibly involve in the interactions with the NR 3′-OH modification and are nearly identical for PARP1 and PARP2, but show large variations in length and sequence among PARP1/PARP2, PARP5a, and PARP10 (Fig. 5a, g). It should be noted that in addition to this extended loop, more subtle differences at the catalytic sites of these PARP enzymes could result in different

activities with **6**. In addition, further work is required to characterize the PAR polymers formed by **6**, including the individual steps of initiation, elongation, and branching for PARP1-catalyzed PARylation with **6** as the substrate.

Additionally, the substrate activity of **6** for recombinant human sirtuin 2 (SIRT2), an NAD$^+$-dependent protein deacetylase, was examined by using a trypsin-coupled fluorescence-based activity assay. It was found that in overnight reactions **6** displays no substrate activity for human SIRT2, suggesting that **6** is not a substrate of SIRT2 and azido substitution at 3′-OH of NR moiety is not tolerated by SIRT2 (Supplementary Fig. 38).

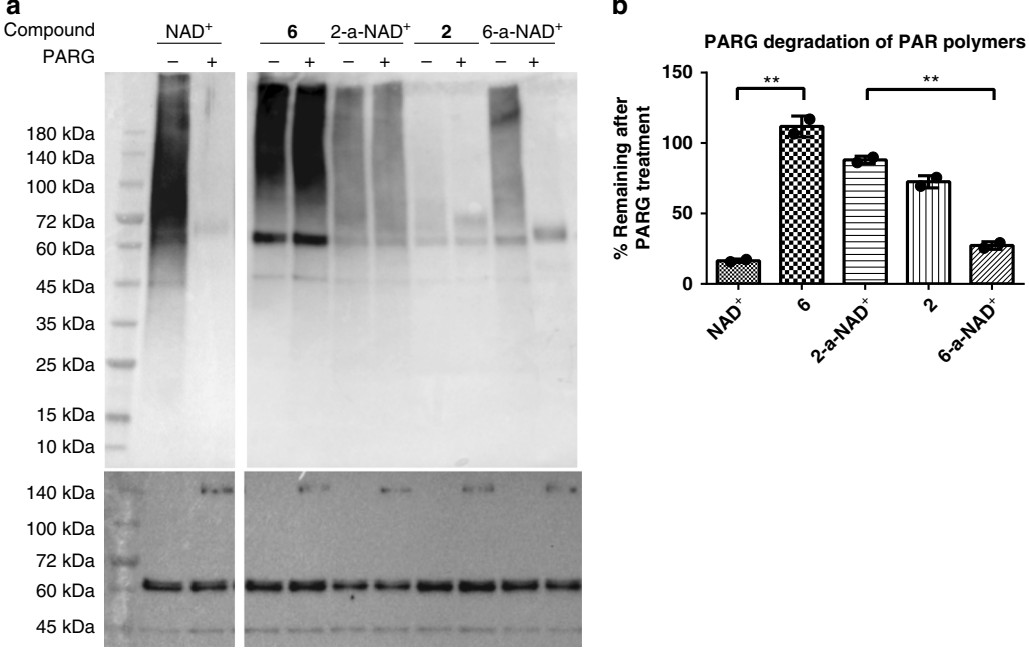

**Fig. 6** PARG-mediated degradation of PAR polymers. **a** Immunoblot analysis of auto-modified PARP1 without and with the addition of human PARG as detected by the Fc-WWE antibody for NAD+-based PAR polymers and the streptavidin-HRP conjugate for NAD+ analogues-derived PAR polymers. PARP1 and PARG loading controls were detected by the anti-His$_6$ antibody. **b** The percentage of remaining PAR polymers post-PARG treatment for auto-modified PARP1 by NAD+ and NAD+ analogues on the basis of densitometric analysis of the PARP1 automodification without and with the addition of human PARG. **P < 0.01 by one-tailed unpaired *t*-test. Error bars represent standard deviation of two replicates. Source data are provided as a Source Data file

This work provides a valuable tool for studying NAD+-dependent PARylation. Notably, the success in generation of this ribose-functionalized NAD+ with robust biological activities potentially enables further development of bi- and multi-functional NAD+ molecules as innovative research tools for investigating cellular signaling and processes mediated by ADP-ribosylation.

## Methods

**General materials and methods.** $^1$H NMR spectra were recorded on an Oxford AM-400 spectrometer for solution in CDCl$_3$, CD$_3$OD or D$_2$O. Coupling constants *J* are shown in Hz. $^{13}$C NMR spectra were recorded on an Oxford AM-400 spectrophotometer (100 MHz) with complete proton decoupling spectrophotometer (CDCl$_3$: 77.0 ppm). Flash column chromatography was performed using 230–400 mesh silica gel (Sigma–Aldrich, St. Louis, MO). For thin-layer chromatography (TLC), silica gel plates (Sigma-Aldrich GF254) were used. HPLC was performed on a Waters 2487 series with C18 Kinetex column (5 μm, 100 Å, 150 × 10.0 mm, from Phenomenex Inc, Torrance, CA). cDNA of human NRK1 (accession number: BC036804) and human NMNAT1 (accession number: BC014943) were purchased from GE Dharmacon (Lafayette, CO). TAMRA-alkyne, azide-fluor 545 (N$_3$-Fluor 545), DAPI, β-NAD+, azide-PEG3-biotin and alkyne-PEG4-biotin, and Calf thymus DNA (activated) were purchased from Sigma–Aldrich. Streptavidin-HRP conjugate was purchased from R&D Systems (Minneapolis, MN). Veliparib was purchased from Selleckchem (Houston, TX). Dithiothreitol (DTT) was purchased from VWR International (Radnor, PA). Trichloroacetic acid (TCA), Pierce Coomassie Plus (Bradford) assay kit, and goat anti-mouse IgG secondary antibody-HRP conjugate (Thermo Fisher Scientific: G-21040) were purchased from Thermo Fisher Scientific (Waltham, MA). Anti-poly-(ADP-ribose) antibody (10H) was purchased from Santa Cruz (Dallas, TX). Recombinant human PARP2 with N-terminal GST tag (product# SRP0193) was purchased from Sigma–Aldrich. HeLa cells were obtained from the American Type Culture Collection (ATCC) (Manassas, VA) and tested negative for mycoplasma. All other reagents were purchased from readily available commercial sources and used without further purification.

**Synthesis and characterization of NAD+ analogues.** The experimental details and results for synthesis of the NAD+ analogues **1–6**, 2-a-NAD+, and 6-a-NAD+ are provided in the Supplementary Information.

**Cloning and generation of NRK1 and NMNAT1.** Protein sequence of human NRK1:MKTFIIGISGVTNSGKTTLAKNLQKHLPNCSVISQDDFFKPESEIET DKNGFLQYDVLEALNMEKMMSAISCWMESARHSVVSTDQESAEEIPILIIEGF

LLFNYKPLDTIWNRSYFLTIPYEECKRRRSTRVYQPPDSPGYFDGHVWPMY LKYRQEMQDITWEVVYLDGTKSEEDLFLQVYEDLIQELAKQKCLQVTAH HHHHHH.

Protein sequence of human NMNAT1:MENSEKTEVVLLACGSFNPITNMH LRLFELAKDYMNGTGRYTVVKGIISPVGDAYKKKGLIPAYHRVIMAELATK NSKWVEVDTWESLQKEWKETLKVLRHHQEKLEASDCDHQQNSPTLERPG RKRKWTETQDSSQKKSLEPKTKAVPKVKLLCGADLLESFAVPNLWKSEDITQ IVANYGLICVTRAGNDAQKFIYESDVLWKHRSNIHVVNEWIANDISSTKIRR ALRRGQSIRYLVPDLVQEYIEKHNLYSSESEDRNAGVILAPLQRNTAEAKT HHHHHH.

The open reading frames (ORFs) of human NRK1 (528 bp) and NMNAT1 (840 bp) with C-terminal His$_6$-tags were amplified through polymerase chain reaction (PCR) using primers of NRK1-Fw/Rv and NMNAT1-Fw/Rv, respectively (Supplementary Table 1). NcoI and XhoI restriction enzyme sites were placed at 5′- and 3′-end of the primers, respectively. After digestion with NcoI and XhoI restriction enzymes, the amplified DNA fragments and backbone of pET-28a (+) vector were purified using gel DNA recovery kits (Zymo Research, CA), followed by ligation using the T4 DNA ligase. All constructed bacterial expression vectors were verified by DNA sequencing (Genewiz LLC, NJ).

To express human NRK1 and NMNAT1, BL21 (DE3) cells were electroporated with the expression constructs and grown in LB Broth containing 50 μg mL$^{-1}$ of kanamycin. After overnight growth, 5 mL of culture was transferred into 1 liter of LB Broth containing 50 μg mL$^{-1}$ of kanamycin and grown in an incubator shaker (250 rpm; Series 25, New Brunswick Scientific, NJ) at 37 °C until the OD$_{600\ nm}$ reached 0.6–0.8. After addition of 0.5 mM isopropyl β-D-1-thiogalactopyranoside (IPTG) to induce protein expression, the bacterial culture was grown in an incubator shaker (250 rpm) overnight at 18 °C, followed by centrifugation at 4550 × *g* (Beckman J6B Centrifuge, JS-4.2 rotor) and resuspension in lysis buffer (20 mM Tris-HCl, pH 8.0, 200 mM NaCl, 20 mM imidazole). Cells were then lysed by passing three times through French Press (GlenMills, NJ) at 25,000 psi and centrifuged at 4 °C for 1 h at 14,000 × *g* (Beckman Coulter centrifuge, JA-17 rotor) to remove debris. The supernatants were filtered using 0.45 μm membranes and loaded on gravity flow columns with 1 mL of Ni-NTA agarose resins (Thermo Fisher Scientific, Waltham, MA). After washing the columns with 15 column volumes of washing buffer (20 mM Tris-HCl, pH 8.0, 200 mM NaCl, 30 mM imidazole), the NRK1 and NMNAT1 were eluted using 15 column volumes of elution buffer (20 mM Tris-HCl, pH 8.0, 200 mM NaCl, 400 mM imidazole). Fractions containing of eluted proteins were combined and dialyzed in the storage buffer (20 mM Tris-HCl, pH 8.0, 300 mM NaCl, 1 mM DTT, 10% glycerol) overnight at 4 °C. After another 6-h dialysis in fresh storage buffer, the purified proteins were concentrated by Amicon centrifugal concentrators (10 kDa cutoff; EMD Millipore, Temecula, CA) and analyzed by SDS-PAGE. Protein concentrations were determined by NanoDrop 2000C spectrophotometer (Thermo

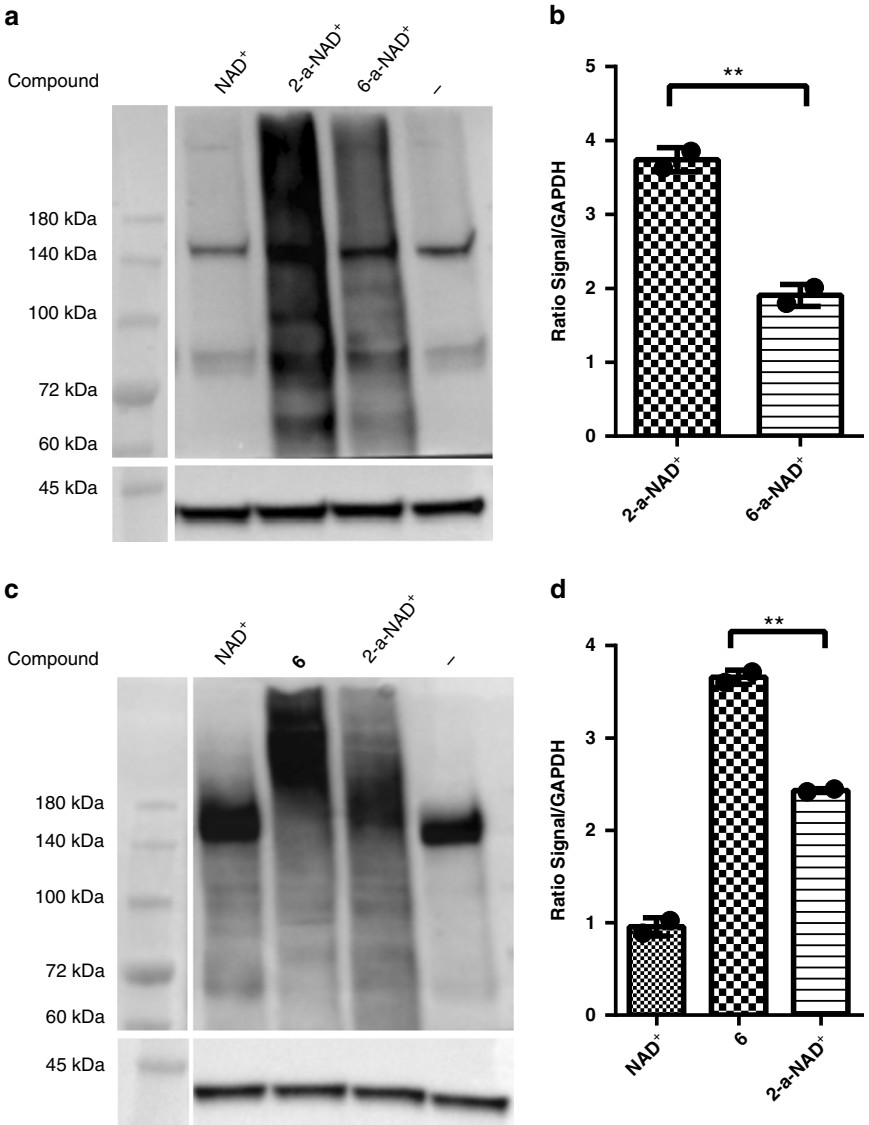

**Fig. 7** Labeling of protein PARylation in cell lysates. **a** Immunoblot analysis of protein PARylation by 2-a-NAD$^+$ and 6-a-NAD$^+$ in HeLa cell lysates as detected by the streptavidin-HRP conjugate. GAPDH as protein loading controls were detected by the anti-GAPDH antibody. **b** Densitometric analysis of protein PARylation normalized to the GAPDH controls. **c** Immunoblot analysis of protein PARylation by NAD$^+$, **6**, and 2-a-NAD$^+$ in HeLa cell lysates as detected by the Fc-WWE antibody. GAPDH as protein loading controls were detected by the anti-GAPDH antibody. **d** Densitometric analysis of protein PARylation normalized to the GAPDH controls. **P < 0.01 by one-tailed unpaired $t$-test. Error bars represent standard deviation of two replicates. Source data are provided as a Source Data file

Fisher Scientific, Waltham, MA) using calculated molecular extinction coefficient (1.537 for NRK1-His$_6$ and 1.592 for NMNAT1-His$_6$). Aliquoted proteins were flash-frozen in liquid nitrogen and stored at −80 °C.

**HPLC of enzymatic conversion of NR and NMN analogues**. NRK1 and NMNAK1 were used to catalyze the conversion processes from NR analogues to NMN analogues and from NMN analogues to NAD$^+$ analogues, respectively. Both enzymes were used in the conversion from NR analogues to NAD$^+$ analogues. All the conversion reactions were performed at room temperature in 100 μL assay solutions containing 50 mM Tris-HCl, pH 7.5, 100 mM NaCl, 20 mM MgCl$_2$, 1 mM DTT, 5 mM ATP, 1 mM NR/NMN analogues and 5 μM purified enzymes. After the indicated incubation time, the reactions were stopped by adding 50% TCA to a final concentration of 10%. After centrifugation at 12,000 × $g$ for 5 min, the supernatants of reaction mixtures were analyzed by reverse-phase HPLC using a semipreparative C18 Kinetex column (5 μm, 100 Å, 150 × 10.0 mm, from Phenomenex Inc, Torrance, CA) with a gradient of methanol (0–40% in 20 min) in water containing 0.1% formic acid.

**Purity analysis of NAD$^+$ analogues 1–6**. Each NAD$^+$ analogue (500 μM) was analyzed separately by reverse-phase HPLC using a semipreparative C18 Kinetex

column (5 μm, 100 Å, 150 × 10.0 mm, from Phenomenex Inc, Torrance, CA) (mobile phase A: 0.1% formic acid (aq); mobile phase B: 0.1% formic acid in methanol; flow rate = 2.0 ml min$^{-1}$; 0–2 min: 0–4% B, 2–4 min: 4–10% B, 4–6 min: 10–20% B, 6–12 min: 20–50% B, 12–14 min: 50–0% B) with detection of UV absorbance at 260 nm. The purity of **1**–**6** was calculated based on the proportion of their corresponding integrated peak areas in the total integrated peak areas.

**Cloning and generation of human PARP1**. cDNA of human PARP1 (accession number: BC037545) was purchased from GE Dharmacon (Lafayette, CO). Full-length human PARP1 with a C-terminal His$_6$-tag was amplified through PCR using primers PX (5′-TGGTGCTCGAGCCACAGGGAGGTCTTAAAATTGAATTTCA GT-3′) and PY (5′-CCCTCTAGAAATAATTTTGTTTAACTTTAAGAAGGAGA TATACCATGGCGGAGTCTTCGGATAAGC-3′) purchased from Integrated DNA Technologies (IDT) (Coralville, IA). The amplified full-length PARP1 and pET-28a (+) vector were digested by XhoI and XbaI restriction enzymes, purified using gel DNA recovery kits (Zymo Research, CA), and then ligated using the T4 DNA ligase. The resulting expression construct for full-length human PARP1 was verified by DNA sequencing (Genewiz LLC, NJ).

The expression and purification of full-length human PARP1 were performed on the basis of a previous report with minor modifications[56]. Briefly, BL21 (DE3) cells were electroporated with the constructed pET-28a vector encoding full-length

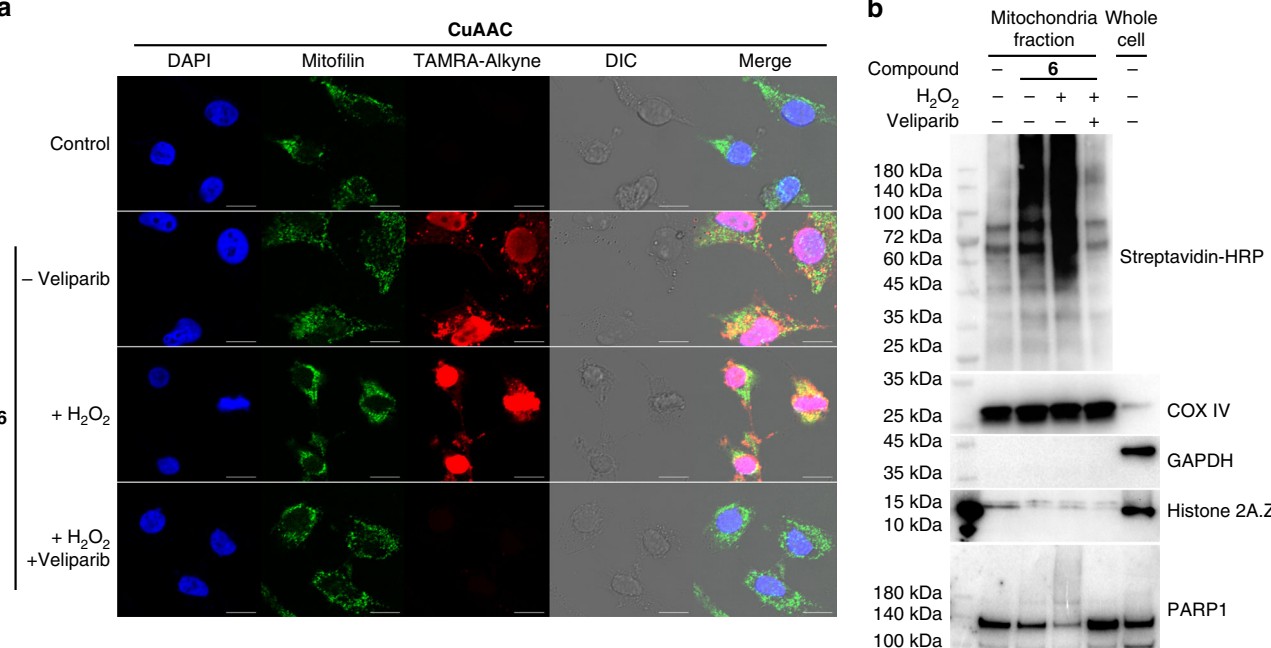

**Fig. 8** Visualization and labeling of mitochondrial protein PARylation by **6**. HeLa cells were pretreated for 20 min in the absence and presence of $H_2O_2$ and 1 μM veliparib. Following permeabilization with 0.025% Triton-X-100, cells were incubated with 100 μM **6** for 45 min, followed by **a** confocal microscopic analysis through fixation, permeabilization, and fluorescent staining by click chemistry, CuAAC: copper(I)-catalyzed azide alkyne cycloaddition, Alexa Fluor 488 conjugated anti-mitofilin antibody, and DAPI. DIC: differential interference contrast. Scale bars: 10 μm, and by **b** immunoblot analysis of mitochondria fraction as detected by a streptavidin-HRP conjugate following biotin conjugation via click chemistry. COX IV as mitochondrial protein loading controls were detected by an anti-COX IV antibody. GAPDH and Histone 2A.Z as cytosol and nuclear protein markers were detected by anti-GAPDH and anti-Histone 2A.Z antibodies, respectively. PARP1 was detected by an anti-PARP1 antibody. Source data are provided as a Source Data file

human PARP1 and grown in LB Broth media containing 50 μg mL$^{-1}$ of kanamycin for overnight. Each liter of LB Broth with 50 μg mL$^{-1}$ of kanamycin was then inoculated with 10 mL of overnight bacterial culture and grown at 37 °C in an incubator shaker (250 rpm; Series 25, New Brunswick Scientific, NJ) until the $OD_{600 nm}$ reached 0.6–0.8, followed by addition of $ZnSO_4$ (final concentration: 0.1 mM). At an $OD_{600 nm}$ of 0.8–1.0, the flasks were removed and incubated on ice for 1 h, followed by induction of protein expression with 0.5 mM IPTG for overnight at 16 °C, centrifugation at 4550 × g (Beckman J6B Centrifuge, JS-4.2 rotor), and resuspension in lysis buffer (25 mM HEPES pH 8.0, 500 mM NaCl, 1 mM PMSF). Cells were lysed by passing a French Press (GlenMills, NJ) for three times at 25,000 psi and then centrifuged at 27,000 × g for 100 min (Beckman Coulter centrifuge, JA-17 rotor) at 4 °C. The collected supernatants were then filtered by passing through 0.45 μm membranes, followed by loading on a gravity flow column with 5 mL of Ni-NTA agarose resins (Thermo Fisher Scientific, Waltham, MA), washing with 50 mL of low-salt wash buffer (25 mM HEPES pH 8.0, 500 mM NaCl, 20 mM imidazole), 50 mL of high-salt wash buffer (25 mM HEPES pH 8.0, 1 M NaCl, 20 mM imidazole) and 50 mL of low-salt wash buffer. Proteins were then eluted with 25 mL elution buffer (25 mM HEPES pH 8.0, 500 mM NaCl, 400 mM imidazole). Twenty-five milliliter no-salt buffer (25 mM Tris pH 7.0, 1 mM EDTA, 0.1 mM DTT) was then added to the eluted proteins and the proteins were loaded onto a 5-mL HiTrap Heparin HP Column by a low-pressure peristaltic pump at a flow rate of at 3 mL min$^{-1}$ (GE Healthcare, Princeton, NJ). Heparin column was placed on an ÄKTA Pure chromatography system to elute PARP1 using a gradient of 0–100% buffer B (50 mM Tris pH 7.0, 1 mM EDTA, 0.1 mM DTT, 1 M NaCl) in buffer A (50 mM Tris pH 7.0, 1 mM EDTA, 0.1 mM DTT, and 250 mM NaCl) at a flow rate of 1 mL min$^{-1}$. PARP1 was eluted starting at 40% buffer B and the collected fractions were combined and spun down to 500 μL using Amicon centrifugal filters with 30 kDa cutoff (EMD Millipore, Temecula, CA). The concentrated proteins were injected on to a size-exclusion chromatography column Superdex 200 Increase 10/300 GL (GE Healthcare, Princeton, NJ) and eluted using gel filtration buffer (25 mM HEPES, pH 8.0, 150 mM NaCl, 1 mM EDTA, 0.1 mM DTT). Purified PARP1 was examined by SDS-PAGE and a NanoDrop 2000C spectrophotometer (Thermo Fisher Scientific, Waltham, MA), then aliquoted and flash-frozen in liquid nitrogen for storage at −80 °C. Calculated molecular extinction coefficient value for human PARP1 with a C-terminal His$_6$-tag is 1.052.

**Auto-PARylation of PARP1 with 1–6.** Auto-PARylation of purified PARP1 was performed at room temperature in 100 μL assay solutions containing 100 mM Tris-

HCl, pH 8.0, 50 mM NaCl, 20 mM $MgCl_2$, 1 mM DTT, 100 ng uL$^{-1}$ activated DNA, 0.4 mM NAD$^+$ or NAD$^+$ analogues **1–6**, and 1 μM purified PARP1 enzyme. PARP1 automodification controls were performed with 100 μM veliparib inhibitor. After indicated incubation time, the reactions were stopped by adding 100 μM veliparib.

The reaction mixtures were further labeled with azide-biotin (for **1–4**) or alkyne-biotin (for **5** and **6**) through copper(I)-catalyzed azide alkyne cycloaddition (CuAAC). Click reactions were performed for 2 h at room temperature in 45 μL volume, which contain 30 μL PARP1 automodification mixtures, 2 mM THPTA, 1 mM $CuSO_4$, 100 μM azide/alkyne-biotin, and 10 mM sodium ascorbate in PBS.

The levels of auto-PARylation were evaluated by immunoblot using an anti-PAR monoclonal antibody (clone: 10H, from Santa Cruz Biotechnology sc-56198; 1:500 dilution) for detection of the formation of ADP-ribose polymer, a streptavidin-HRP conjugate (R&D Systems: DY998; 1:200 dilution) for detection of the biotinylated PARP1 via click chemistry, and an anti-His$_6$ antibody (clone: HIS. H8, from Thermo Fisher Scientific MA1-21315; 1:2000 dilution) for detection of PARP1 as loading control. The goat anti-mouse IgG antibody-HRP conjugate (Thermo Fisher Scientific: G-21040; 1:3000 dilution) was used as the secondary antibody for the anti-PAR antibody and anti-His$_6$ antibody. Uncropped and unprocessed scans of the blots are provided as a Source Data file.

**HPLC-based kinetic assays.** Auto-PARylation of PARP1 was carried out in 80 μL assay solutions (30 mM HEPES, pH 8.0, 5 mM $MgCl_2$, 5 mM $CaCl_2$, 1 mM DTT, 100 ng μL$^{-1}$ of activated DNA, 100 ng μL$^{-1}$ BSA) containing varied concentrations of NAD$^+$ (50, 100, 250, 450, 600 and 750 μM) or NAD$^+$ analogues (**2**, **6**, 6-a-NAD$^+$, and 2-a-NAD$^+$) at 30 °C with purified PARP1 enzymes. The reactions were quenched at different time points (NAD$^+$: 0, 2.5, 5, 10, 15, and 20 min; **2**: 0, 60, 120, 180, 240 and 300 min; **6**: 0, 5, 10, 15, 20 and 30 min; 6-a-NAD$^+$ and 2-a-NAD$^+$: 0, 10, 20, 30, and 40 min) using 20% ice-cold TCA. After centrifugation, the reaction mixtures were analyzed by reverse-phase HPLC using a semi-preparative C18 Kinetex column (5 μm, 100 Å, 150 × 10.0 mm, from Phenomenex Inc, Torrance, CA) (mobile phase A: 0.1% formic acid (aq); mobile phase B: 0.1% formic acid in acetonitrile; flow rate = 2.0 mL min$^{-1}$; 0–8 min: 0% B, 8–12 min: 0–50% B, 12–13 min: 50–2.5% B, 13–18 min: 2.5–40% B, 18–20 min: 40–0% B) with detection of UV absorbance at 260 nm. The retention times for NAD$^+$, **2**, **6**, 6-a-NAD$^+$ and 2-a-NAD$^+$ were 13.0, 13.6, 13.6, 14.0 and 13.8 min, while for ADPr, ADPr2, ADPr6, ADPr6-a-NAD$^+$, and ADPr2-a-NAD$^+$, the retention times were 6.3, 12.9, 9.0, 13.0, and 14.2 min. Standard curves for NAD$^+$ and NAD$^+$ analogues together with ADPr and ADPr analogues were constructed by linear

correlations of concentrations and corresponding integrated peak areas. NADase reaction rates were determined based on the increase in peak areas of the assigned peaks of ADPr and ADPr analogues, while reaction rates for PARP activities were measured based on the decrease in peak areas of the assigned peaks of $NAD^+$ and $NAD^+$ analogues excluding NADase activity. Kinetic parameters were determined by fitting data to the Michaelis-Menten model implemented in GraphPad Prism (La Jolla, CA).

**Cellular PARylation assays**. Detection of PARylation activities in oxidatively stressed cells was performed by following a previously established assay with minor modifications[27]. HeLa cells were first incubated with 500 μM $H_2O_2$ in PBS for 20 min. As negative controls, cells were pre-incubated for 2 h with 1 μM veliparib prior to the additions of $H_2O_2$, followed by washing once with PBS. For pre-permeabilized groups, cells were incubated with reaction buffer (56 mM HEPES, 28 mM KCl, 28 mM NaCl, 2 mM $MgCl_2$, pH 8.0) complemented with 0.01% Triton-X-100, and 100 μM $NAD^+$ analogues for 15 min at 4 °C. After removal of the reaction buffer with 0.01% Triton-X-100, cells were washed with PBS and incubated with reaction buffer and 100 μM/200 μM $NAD^+$ analogues in the absence of Triton-X-100 for 45 min at 37 °C, followed by washing three times with PBS. For non-permeabilized control groups, cells were incubated with reaction buffer (56 mM HEPES, 28 mM KCl, 28 mM NaCl, 2 mM $MgCl_2$, pH 8.0) and 100 μM/200 μM $NAD^+$ analogues for 1 h at 37 °C, followed by washing three times with PBS. For control experiments, veliparib was added in all experimental steps.

For confocal imaging analysis, cells were fixed with 4% para-formaldeyhde (PFA) for 20 min at room temperature, followed by washing three times with PBS, permeabilization with 0.5% Triton-X-100 in PBS for 20 min at room temperature, and washing twice with 3% BSA/PBS. CuAAC reactions were performed with 10 mM THPTA, 5 mM $CuSO_4$, 5 μM alkyne-TAMRA or azide-Fluor 545, 50 mM sodium ascorbate in PBS at room temperature for one hour, followed by washing three times with PBS. Cell nuclei were stained with DAPI (1 μg mL$^{-1}$ in PBS) for 20 min at room temperature, followed by washing three times with PBS and confocal microscopic analysis using a Leica SP8 confocal laser scanning microscope (Leica Microsystems Inc., Buffalo Grove, IL) equipped with a 63×, 1.4 NA PLAPO oil immersion objective lens. DAPI, TAMRA, and Fluor-545 were excited at 405 nm, 557 nm, and 545 nm, respectively. Images were processed using LAS X software (Leica Microsystems Inc., Buffalo Grove, IL).

For immunoblot analysis, cells were harvested and lysed on ice for 15 min using M-PER Mammalian Protein Extraction Reagent (Thermo Fisher Scientific, Waltham, MA) with protease inhibitors. After centrifugation at 12,000 × g for 10 min at 4 °C, proteins in supernatant were labeled with biotin for 2 h at room temperature by performing CuAAC reactions in 45 μL click solution, which contained 30 μL cell lysates, 2 mM THPTA, 1 mM $CuSO_4$, 100 μM azide/alkyne-biotin, and 10 mM sodium ascorbate in PBS buffer. The reaction mixtures were analyzed by immunoblot using a streptavidin-HRP conjugate (R&D Systems: DY998; 1:200 dilution) for biotinylated cellular proteins and an anti-GAPDH antibody (clone: GA1R, from Thermo Fisher Scientific MA5–15738; 1:3000 dilution) for detecting the levels of GAPDH as loading controls. The goat anti-mouse IgG antibody-HRP conjugate (Thermo Fisher Scientific: G-21040; 1:3000 dilution) was used as the secondary antibody for the anti-GAPDH antibody. Uncropped and unprocessed scans of the blots are provided as a Source Data file.

The effect of transient permeabilization on cell viability was evaluated with trypan blue and (3-(4,5-dimethylthiazol-2-yl)-2,5-diphenyltetrazolium bromide) (MTT), respectively. HeLa cells were incubated with reaction buffer (56 mM HEPES, 28 mM KCl, 28 mM NaCl, 2 mM $MgCl_2$, pH 8.0) complemented with 0.01% Triton-X-100 for 15 min at 4 °C. After removal of the reaction buffer with 0.01% Triton-X-100, cells were washed with PBS and incubated with fresh medium at 37 °C. HeLa cells incubated with reaction buffer or medium in the same condition were included as controls. Following 1-h incubation, cell viability was determined by the trypan blue method. The numbers of blue staining cells and the numbers of total cells were examined by optical microscopy. The percentage of viable cells was calculated as: %viable cells = [1.00 − (number of blue cells/number of total cells)] × 100. Following 24-h incubation, cell viability was determined by MTT assays. The cells were incubated with MTT solution for 2 h to allow viable cells to reduce the yellow tetrazolium salt (MTT) into dark blue formazan crystals. 100 μL of lysis buffer (20% SDS in 50% dimethylformamide, pH 4.7) was then added to wells and incubated for 2 h at 37 °C. The absorbance was measured at 570 nm using a BioTek Synergy H1 Hybrid Multi-Mode Microplate reader. Percent viability was calculated as: %cell viability = [(absorbance$_{experimental}$ − absorbance$_{spontaneous\ average}$)/(absorbance$_{maximum\ viability\ average}$ − absorbance$_{spontaneous\ average}$)] × 100.

To detect PARylation activities in mitochondria, HeLa cells were treated as described above except that 0.025% Triton-X-100 was used in the permeabilization process.

For confocal imaging analysis, cells were blocked with 3% BSA/PBS for 30 min at room temperature after the CuAAC reactions as described above, and then incubated with Alexa Fluor 488 conjugated anti-mitofilin antibody (1:200 in 3% BSA/PBS; clone: MM#7A-2E4AD5, from Thermo Fisher Scientific MA4-56400-A488) for 1 h at room temperature, followed by washing three times with PBS. Cell nuclei were stained with DAPI (1 μg mL$^{-1}$ in PBS) for 20 min at room temperature, followed by washing three times with PBS and confocal microscopic

analysis using a Leica SP8 confocal laser scanning microscope (Leica Microsystems Inc., Buffalo Grove, IL) equipped with a 63×, 1.4 NA PLAPO oil immersion objective lens. DAPI, TAMRA, and Fluor-488 were excited at 405 nm, 557 nm, and 488 nm, respectively. Images were processed using LAS X software (Leica Microsystems Inc., Buffalo Grove, IL).

For immunoblot analysis, cells were harvested, and mitochondria fractions were isolated using Qproteome Mitochondria Isolation Kit (from QIAGEN, Cat No.: 37612) by following the manufacturer's protocol. Isolated mitochondria fractions were lysed on ice for 15 min using lysis buffer (25 mM Tris-HCl pH 7.5, 50 mM NaCl, 10% glycerol, 1% Nonidet P-40) with protease inhibitors. After centrifugation at 12,000 × g for 10 min at 4 °C, proteins in supernatants were labeled with biotin for 2 h at room temperature by performing CuAAC reactions as described above. The reaction mixtures were analyzed by immunoblots using a streptavidin-HRP conjugate (R&D Systems: DY998; 1:200 dilution) for biotinylated proteins, a monoclonal anti-COX IV antibody (clone: 3E11, from Cell Signaling Technology #4844; 1:3000 dilution) for COX IV as mitochondrial protein loading controls, a monoclonal anti-GAPDH antibody (clone: GA1R, from Thermo Fisher Scientific MA5-15738; 1:3000 dilution) for GAPDH as a cytoplasmic protein marker, a polyclonal anti-Histone 2A.Z antibody (from Cell Signaling Technology product #2718S; 1:500 dilution) for Histone 2A.Z as a nuclear protein marker, and a monoclonal anti-PARP1 antibody (clone: 46D11, from Cell Signaling Technology #2718S; 1:2000 dilution) for PARP1. The goat anti-mouse IgG antibody-HRP conjugate (Thermo Fisher Scientific: G-21040; 1:3000 dilution) was used as the secondary antibody for the anti-GAPDH antibody and the goat anti-rabbit IgG antibody-HRP conjugate (Thermo Fisher Scientific: G-21234; 1:3000 dilution) was used as the secondary antibody for the anti-COX IV antibody, anti-Histone 2A.Z antibody and anti-PARP1 antibody. Uncropped and unprocessed scans of the blots are provided as a Source Data file.

**Generation of human PARP5a and PARP10 catalytic domains**. cDNAs of human PARP10 (accession number: BC144235) and PARP5a (GE Dharmacon: MHS6278-202806376) were used as templates for polymerase chain reaction to amplify the catalytic domains (residues 809-1017 and residues 1093-1327 for PARP10 and PARP5a, respectively) and add a His$_6$-tag at the N-terminus or C-terminus. Primers #1 (5′- GCCGGATCTCAGTGGTGGTGGTGGTGGTGC TCGAGTTATTACCCAGAGGGGTCGTCGGG-3′) and #2 (5′- AAGGAGATAT ACCATGCACCATCATCATCATCATATTGAGGGTCGCGCGGGGCAGACGC TGAAGG-3′) were used to amplify PARP10, and primers #3 (5′- GTGGTGCTC GAGTTATTAATGATGATGATGATGGTGGGTCTTCTGCTCTGCGGCTGTT G-3′) and #4 (5′-TTAACTTTAAGAAGGAGATATACCATGACCAATCCTTAT TTGACTTTTCACTGTGTTAATCAG-3′) were used to amplify PARP5a. PCR products were verified by agarose gel electrophoresis and an additional round of PCRs were completed using primers #5 (5′- TTCCCCTCTAGAAATAATTT TGTTTAACTTTAAGAAGGAGATATACCATGCACCATCATCATC-3′) and #1 to extend the PCR product of PARP10 and primers #6 (5′- TTCCCCTCTAGA AATAATTTTGTTTAACTTTAAGAAGGAGATATACCATGACCAATCCT-3′) and #3 for PARP5a. The PCR products were purified by agarose gel electrophoresis and digested using XbaI and XhoI (New England Biolabs) for 1 h at 37 °C. pET-28a (+) plasmids were digested overnight using XbaI and XhoI at 37 °C and cut vector was purified from agarose gel electrophoresis. Digested PCR fragments and vector were ligated using T4 DNA Ligase (New England Biolabs) for 1 h at room temperature prior to transformation into DH10B electrocompetent cells.

Sequence-verified plasmids were then transformed into BL21 (DE3) cells for bacterial expression and purification. 5 mL of LB broth with 50 μg mL$^{-1}$ kanamycin were inoculated with transformed BL21 (DE3) cells with the plasmids encoding PARP10 and PARP5a catalytic domains and grown overnight at 37 °C at 250 rpm. Each overnight culture was inoculated into 1 L of LB broth with 50 μg mL$^{-1}$ kanamycin and grown to OD$_{600\ nm}$ = 0.8 at 37 °C at 250 rpm. Protein expression was induced using a final concentration of 500 μM IPTG and grown overnight at 22 °C at 250 rpm. Cells were harvested by centrifuging for 30 min at 2,700 × g and discarding the supernatant. Cell pellets were resuspended in 30 mL equilibrium buffer (20 mM Tris pH 7.5, 500 mM NaCl, 20 mM imidazole) with 1 mg mL$^{-1}$ lysozyme (Amresco Inc.), 10 μg mL$^{-1}$ DNase I (Roche), 0.1 mM $MgCl_2$ and 0.1 mM phenylmethylsulfonyl fluoride (Chem-Impex Int'l. Inc.). Cells were lysed by running cells through a French press (Glen Mills Inc.) at 25,000 psi three times. Cell debris was spun down at 27,000 × g. 1 mL of Ni-NTA agarose resin (Thermo Fisher Scientific) was loaded onto a gravity flow column and washed with 15 column volume (CV) of water and 15 CV of equilibrium buffer before running cell lysate through. The column was then washed with 15 CV of equilibrium buffer and 15 CV of wash buffer (20 mM Tris pH 7.5, 500 mM NaCl, 30 mM imidazole). Bound proteins were eluted with 15 CV elution buffer (20 mM Tris pH 7.5, 500 mM NaCl, 400 mM imidazole). Proteins were concentrated with 10 kDa MWCO ultra-15 centrifugal filter units (Amicon) before running through an AKTA FPLC (GE Healthcare) with a SuperDex 75 10/300 column (GE Healthcare) preequilibrated with storage buffer (20 mM Tris pH 7.5, 300 mM NaCl, 10% glycerol, 1 mM DTT). An isocratic gradient with a flow rate of 0.5 mL min$^{-1}$ was used to separate 0.5 mL fractions, which were run on a PAGE gel to identify fractions containing the desired proteins. Fractions were combined and concentrated by the centrifugal filter units. Protein concentrations were examined by SDS-PAGE and spectrophotometry, analyzing the wavelength at 280 nm and

using extinction coefficients of 0.762 and 0.753 for PARP10 and PARP5a, respectively.

**Substrate activities of NAD+ analogues for PARPs**. Auto-PARylation of purified PARP1 was performed at room temperature in 100 μL of assay solutions containing 100 mM Tris-HCl, pH 8.0, 50 mM NaCl, 20 mM MgCl₂, 0.2 mM DTT, 100 ng μL⁻¹ activated DNA, 0.4 mM NAD+ or NAD+ analogues **2**, **6**, 6-a-NAD+, 2-a-NAD+, and 1 μM purified PARP1 enzyme. PARP1 automodification controls were performed with 100 μM olaparib inhibitor. After indicated incubation time, the reactions were stopped by adding 100 μM olaparib.

Auto-PARylation of recombinant human PARP2 was performed at room temperature in 10 μL of assay solutions containing 100 mM Tris-HCl, pH 8.0, 50 mM NaCl, 20 mM MgCl₂, 0.2 mM DTT, 100 ng μL⁻¹ activated DNA, 0.1 mM NAD+ or NAD+ analogues **2**, **6**, 6-a-NAD+, 2-a-NAD+, and 200 ng purified PARP2 enzyme. PARP2 automodification controls were performed with 100 μM olaparib inhibitor. After indicated incubation time, the reactions were stopped by adding 100 μM olaparib.

Auto-PARylation of the purified PARP5a catalytic domain was performed at room temperature in 100 μL of assay solutions containing 100 mM Tris-HCl, pH 8.0, 200 mM NaCl, 20 mM MgCl₂, 0.2 mM DTT, 0.4 mM NAD+ or NAD+ analogues **2**, **6**, 6-a-NAD+, 2-a-NAD+, and 10 μM purified PARP5a catalytic domain. PARP5a automodification controls were performed with 100 μM olaparib inhibitor. After indicated incubation time, the reactions were stopped by adding 100 μM olaparib.

Auto-MARylation of the purified PARP10 catalytic domain was performed at room temperature in 100 μL of assay solutions containing 100 mM Tris-HCl, pH 8.0, 200 mM NaCl, 20 mM MgCl₂, 0.2 mM DTT, 0.4 mM NAD+ or NAD+ analogues **2**, **6**, 6-a-NAD+, 2-a-NAD+, and 20 μM purified PARP10 catalytic domain. PARP10 automodification controls were performed with 100 μM olaparib inhibitor. After indicated incubation time, the reactions were stopped by adding 100 μM olaparib.

The reaction mixtures were further labeled with azide-biotin (for NAD+, **2**, 6-a-NAD+, and 2-a-NAD+) or alkyne-biotin (for NAD+ and **6**) through copper(I)-catalyzed azide alkyne cycloaddition (CuAAC). Click reactions were performed for 2 h at room temperature in 45 μL volume, which contained 30 μL of PARP1/2/5a/10 automodification mixtures, 2 mM THPTA, 1 mM CuSO₄, 100 μM azide/alkyne-biotin, and 10 mM sodium ascorbate in PBS.

The levels of automodification were evaluated by immunoblot using a streptavidin-HRP conjugate (R&D Systems: DY998; 1:200 dilution) for detection of the biotinylated PARP1/2/5a/10 via click chemistry, an anti-His₆ antibody (clone: HIS.H8, from Thermo Fisher Scientific MA1-21315: 1:2000 dilution) for detection of PARP1/5a/10 as loading controls, and an anti-PARP2 antibody (clone: F-3, from Santa Cruz Biotechnology sc-393310; 1:200 dilution) for detection of PARP2 as loading controls. The goat anti-mouse IgG antibody-HRP conjugate (Thermo Fisher Scientific: G-21040; 1:3000 dilution) was used as the secondary antibody for the anti-His₆ antibody and anti-PARP2 antibody. Uncropped and unprocessed scans of the blots are provided as a Source Data file.

**PARG-mediated degradation of ADP-ribose polymers**. Auto-PARylation of PARP1 was performed at 30 °C overnight using 3 μM purified PARP1 and 400 μM NAD+ or NAD+ analogues in a reaction buffer containing 100 mM Tris-HCl pH 8.0, 10 mM MgCl₂ 50 mM NaCl, 1 mM DTT, and 100 ng μL⁻¹ activated DNA. 10 ng of human PARG (Sigma–Aldrich: SRP8023) was added to 300 ng of PARP1 auto-modified with NAD+ or NAD+ analogues in PBS in a total volume of 10 μL for 30 min at room temperature to degrade ADP-ribose polymers. Two hundred nanogram of the reaction was quenched with LDS loading dye (Thermo Fisher Scientific: NP0008) and 100 ng was used for CuAAC before being quenched with LDS loading dye. One hundred nanogram of each quenched reaction was run on precast PAGE gels for detection by an anti-His₆ antibody, Fc-WWE antibody (EMD Millipore: MABE1031), or streptavidin-HRP conjugate. Proteins were transferred onto PVDF membrane and blocking was done with 5% nonfat milk in PBS with 0.05% Tween-20 (PBST) for 1 h at room temperature for detection by the Fc-WWE antibody and anti-His₆ antibody. 3% BSA in PBST was used for blocking the membrane for 1 h at room temperature for detection by the streptavidin-HRP conjugate. The Fc-WWE antibody was used at 1:5000 in PBST, anti-His₆ antibody was used at 1:3000 in PBST, and streptavidin-HRP conjugate was used at 1:200 in PBS for 1 h at room temperature. Anti-rabbit antibody-HRP conjugate (secondary antibody of System Biosciences: ExoAb-CD63A-1; 1:3000 dilution) was used as the secondary antibody for the Fc-WWE antibody and anti-mouse antibody-HRP conjugate (Thermo Fisher Scientific: G-21040; 1:3000 dilution) was used as the secondary antibody for the anti-His₆ antibody for 1 h. Detection was done using SuperSignal West Pico PLUS Chemiluminescent Substrate (Thermo Fisher Scientific: 34580). Images were analyzed and quantified using ImageJ (https://imagej.nih.gov/ij/index.html)[57].

**Binding of WWE domain to auto-modified PARP1**. Auto-PARylation of PARP1 was performed at 30 °C overnight using 3 μM purified PARP1 and 400 μM NAD+ or NAD+ analogues **6** and 2-a-NAD+ in a reaction buffer containing 100 mM

Tris-HCl pH 8.0, 10 mM MgCl₂ 50 mM NaCl, 1 mM DTT, and 100 ng μL⁻¹ activated DNA. Ten nanogram of PARP1 auto-modified with NAD+ and NAD+ analogues **6** and 2-a-NAD+ were diluted into 100 μL of PBS and plated into 96-well high-binding plates (Greiner Bio-One: 655077) and incubated overnight at room temperature. Following three washes using 400 μL of PBST, wells were blocked with 300 μL 3% BSA in PBS for 2 h at room temperature. This was followed by another three washes with PBST before adding the Fc-WWE antibody at 1:5000 in PBST supplemented with 0.1% BSA for 2 h at room temperature. An anti-rabbit antibody-HRP conjugate was used as the secondary antibody at a dilution factor of 1:3000 in PBST with 0.1% BSA for 1 h at room temperature. Detection was done using QuantBlu Fluorogenic Peroxidase Detection (Thermo Fisher Scientific: 15169) with an excitation wavelength of 325 nm and an absorption wavelength of 420 nm. Results and graphs were generated using Graphpad Prism (GraphPad Software, La Jolla, CA).

**Preparation of HeLa cell lysates**. HeLa cells were grown in T25 flasks to 90% confluency and then collected using trypsin to detach the cells from the flasks before spinning down and washing the cell pellet with PBS. Two hundred microliter of lysis buffer (25 mM Tris-HCl pH 7.5, 50 mM NaCl, 10% glycerol, 1% Nonidet P-40 (VWR: 97064-734), and Halt Protease Inhibitor Cocktail (Thermo Fisher Scientific: 78430)) was used to resuspend the cell pellets. Cells were shaken for 10 min at room temperature before spinning down cellular debris at 14,000 × g at 4 °C for 15 min. Protein concentrations of the cell lysates were measured using Bradford reagent (Thermo Fisher Scientific: 23236).

**Labeling of protein PARylation in cell lysates**. Two hundred micromolar of NAD+ or NAD+ analogue was added to 10 μg of lysate diluted in PBS to a final volume of 10 μL. Reactions were incubated at 30 °C for 1 h. Five microgram of lysate was quenched by the addition of LDS loading dye and analyzed on PAGE gels; the remaining 5 μg was subjected to CuAAC before quenching with LDS dye and run on PAGE gels. Proteins were transferred onto PVDF membranes and blocked with 5% nonfat milk or 3% BSA. Protein PARylation was detected using the streptavidin-HRP conjugate for 2-a-NAD+ and 6-a-NAD+ or Fc-WWE antibody for **6** and 2-a-NAD+. The anti-rabbit antibody-HRP conjugate was used as the secondary antibody for the Fc-WWE antibody. The anti-GAPDH antibody was used for a loading control at 1:3000 in PBST, and the anti-mouse antibody-HRP conjugate was used at 1:3000 as the secondary antibody. Detection was done as described above. Images were analyzed by ImageJ.

**Substrate activities of NAD+ and 6 for human SIRT2**. 0.5 mM NAD+ or **6** was incubated with 0.25 mM acetylated peptide substrate Ac-Arg-Gly-Lys(Ac)-AMC (R&D System, MN) and 45.4 nM recombinant human SIRT2 (R&D System, MN) in buffer containing 25 mM Tris pH 8.0, 150 mM NaCl, 1 mM DTT for overnight at room temperature. Reactions without SIRT2 were included as controls. Following the overnight reactions, 50 nM bovine trypsin (Sigma-Aldrich, MO) was added to reaction mixtures for cleaving the deacetylated peptide substrate and releasing fluorescent 7-amino-4-methylcoumarin (AMC) group (excitation 380 nm; emission 460 nm). The reactions were monitored by using Synergy H1 microplate reader (Biotek, VT) on the basis of fluorescence intensity.

**Statistical analysis**. One-tailed unpaired $t$ tests were performed for comparison between two groups. A $P < 0.05$ was considered statistically significant. Significance of finding was defined as: ns = not significant, $P > 0.05$; *$P < 0.05$; **$P < 0.01$; and ***$P < 0.001$. Data are shown as mean ± SD. All statistical analyses were calculated using GraphPad Prism (GraphPad Software, CA).

**Reporting summary**. Further information on research design is available in the Nature Research Reporting Summary linked to this article.

## Data availability
The authors confirm that the data supporting the findings of this study are available from the corresponding author upon request. The source data underlying Figs. 3, 4b, 5c–f, 6, 7 and 8b and Supplementary Figs. 22, 31–33, 34b–36b, 37, and 38 are provided as a Source Data file.

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

## Acknowledgements

The authors thank the Cell and Tissue Imaging Core of USC Research Center for Liver Disease (supported by NIH P30DK048522) for providing microscopy services. This work was supported by University of Southern California School of Pharmacy Start-Up Fund for New Faculty, Sharon L. Cockrell Cancer Research Fund, The V Foundation for Cancer Research V Scholar Grant V2016-021 (to Y.Z.), UCLA Eli and Edythe Broad Center of Regenerative Medicine and Stem Cell Research Hal Gaba Director's Fund for Cancer Stem Cell Research (to N.M.E.), and University of Southern California Research Center for Liver Diseases Pilot Grant P30DK048522 (to Y.Z.).

## Author contributions

X.N.Z., Q.C. and Y.Z. designed research; X.N.Z., Q.C., J.C., A.T.L., Y.L., Z.D. and N.M.E. performed research; H.P. and S.G.L. provided resources and critical insights; X.N.Z., Q.C., J.C., A.T.L., Z.D. and Y.Z. analyzed data; and X.N.Z., Q.C. and Y.Z. wrote the manuscript.

## Additional information

**Competing interests:** The authors declare no competing interests.

