## [Peer Review File · Nature Communications]

Reviewers' comments:

Reviewer #1 (Remarks to the Author):

I previously reviewed the manuscript for another journal and I recommended for publication after one round of revision that addressed my initial comments. So my comments will be brief here and I again support the publication at Nature Communication.

The manuscript describes the development of new NAD⁺ analogs that can be used to label the substrate proteins of PARPs. The authors have demonstrated that the new probes have certain improved properties compared to existing ones. Given the important functions of PARPs in various biological pathways, I believe such probes will be useful for the elucidating the functions of PARPs. Furthermore, the chemo-enzymatic methods developed here will also be useful for the synthesis of other NAD⁺ analogs. For these two reasons, I support the publication of the manuscript.

Hening Lin

Reviewer #2 (Remarks to the Author):

NCOMMS-19-06435

A Ribose-Functionalized NAD⁺ with Unexpected High Activity and Selectivity for Protein Poly-ADP-Ribosylation

Zhang et al generate NAD⁺ derivatives with functional groups added to the nicotinamide ribose. They accomplish this by using the enzyme NRK1 to phosphorylate modified variants of nicotinamide ribose as starting materials, and then the enzyme NMNAT to join the modified nicotinamide mononucleotide (NMN) variants with ATP to produce the NAD⁺ derivatives. They develop an efficient method for producing and purifying these NAD⁺ derivatives, and then test whether and how efficiently they can serve as substrates for the enzyme PARP-1, relative to the native substrate NAD⁺. One of the derivatives, compound 6, is utilized by the enzyme PARP1 with an efficiency similar to that of NAD⁺. An interesting feature of the poly(ADP-ribose) made from this compound is that it appears to be resistant to degradation by PARG, poly(ADP-ribose) glycohydrolase, and could thus potentially serve as an interesting tool for investigating the cellular consequences persistent poly(ADP-ribose) levels. Another interesting feature is that compound 6 appears to be poorly utilized by PARP5a and PARP10. There are limitations to the utility of compound 6 in that it appears to require permeabilization of cells in order to be delivered. The approach is interesting and thoroughly done, and the derivatives could potentially answer some interesting biological questions. In this regard, the study is mostly tool development and does not really push the boundaries of our knowledge of PARP enzymology or cell biology. An example of how this new technology can be used to improve our knowledge would add an important element to the study and broaden the importance of the work.

Reviewer #3 (Remarks to the Author):

As the first point in review, it is important to state that this paper offers a solution to a long standing problem in poly(ADP-ribose)polymerase (PARP) research--the inability to find NAD analogs that are efficient PARP substrates. The data presented in this paper supports the conclusion that azido substitution for the 3'-OH of the nicotinamide riboside of NAD is recognized by PARP, and is polymerized into an ADP-ribose polymer. This observation represents an advance, and the synthesis and properties of this novel PARP substrate, justifies publication in NATURE Communications. The chemo-enzymatic synthesis of this NAD substrate is an accomplishment, as is its chemical synthesis that is reported in the Supplementary Information section.

This study does not isolate and characterize the polymers that are produced by PARP using 3-N₃-NAD as a substrate. Neither does it distinguish between the analogs relative rate of reaction for the initiation, the subsequent polymerization and the branching steps. It is therefore premature to

conclude that high molecular branched polymers of similar structure to that formed using NAD as a substrate are the result when using the 3-N₃-NAD. Clarification will require additional study, and in this communication the authors should simply and clearly point out that more work will be required to characterize the products of the reaction with the new substrate 3-N₃-NAD.

There are several minor points that I believe that authors should be asked to address prior to publication:

1. On page 5-6, yields of 68% are reported for the enzymatic conversion of NR6 to the NAD analog 6. It will be of interest to a reader to know if this refers to an isolated yield of 6 or to a yield calculated on the basis of HPLC analysis at the conclusion of the reaction. The authors should clarify the reporting of this result. It will also be of interest to know the maximum quantities of 6 that have been produced using this enzymatic synthesis.

2. Page 7 line 162-163. Lack of substrate activity for compound 5 could alternately be due to electronic properties of 5. The electron withdrawing 2-azide adjacent to the pyridinium-leaving group is predicted to decrease the rate of pyridine-ribose bond cleavage significantly. In a previous study [A.L.Hanlon et. al., JACS (1994) 116, 12087-12088], 2'-azido-NAD was shown to be resistant to both NAD glycohydrolase catalyzed and to chemical hydrolysis. The 2'-azide was hydrolyzed by the NAD glycohydrolase approximately 10,000-fold more slowly than was NAD. Compound 5 could therefore be a slowly cleaved or a non-cleavable substrate analog. Incidentally, 2-N₃-NAD was first reported in this study.

3. Page 14, lines 297-298. The experiment that the authors report demonstrates that 6 is not a substrate for SIRT2, not that there is an essential role for the nicotinamide ribotide 3'-OH in NAD analog 6. An azide group is bigger and electronically different from an -OH, and the azide simply might not fit or might constrain that analog into a different confirmation. We can only conclude that compound 6 is not a substrate and that substitution of an -OH for an -N₃ is not tolerated by this enzyme.

4. It is important that the authors define the purity of the compounds that are tested in this study. It is expected that tested compounds will be highly pure. Therefore I would recommend the addition of a table or a paragraph to Methods that describes the purity of NAD analogs 1-6 and the method or methods that were used to assess the tested compounds purity. I expect (from the Supplementary Information) that the method will be analytical HPLC, but the purity of the compounds tested is never explicitly defined in the description of their synthesis.

5. The synthesis of the six NAD analogs described in this study begins with the construction of substituted ribose derivatives, the formation of a 1-bromo-ribose derivative, and the displacement of the 1-bromide with nicotinamide to form the nicotinamide-ribose derivative. The stereochemistry of the reaction product must be defined, and it's additionally important to insure that the NAD derivatives that are eventually synthesized contain the expected nicotinamide beta-riboside. Normally the beta-linkage is formed when the bromo-ribose has a 2-benzoyl substituent, but this is not the case for compound 5-7 and NR-5 where a 2-azide is present. In this case and for 1-7 and 3-7 (containing 2-ether substituents) how was the stereochemistry of the products assigned, and how was it proved that the products are the beta-anomers as drawn?

Reviewer 1

The manuscript describes the development of new NAD⁺ analogs that can be used to label the substrate proteins of PARPs. The authors have demonstrated that the new probes have certain improved properties compared to existing ones. Given the important functions of PARPs in various biological pathways, I believe such probes will be useful for the elucidating the functions of PARPs. Furthermore, the chemo-enzymatic methods developed here will also be useful for the synthesis of other NAD⁺ analogs. For these two reasons, I support the publication of the manuscript.

We do appreciate the reviewer's comments.

Reviewer 2

.... There are limitations to the utility of compound 6 in that it appears to require permeabilization of cells in order to be delivered. The approach is interesting and thoroughly done, and the derivatives could potentially answer some interesting biological questions. In this regard, the study is mostly tool development and does not really push the boundaries of our knowledge of PARP enzymology or cell biology. An example of how this new technology can be used to improve our knowledge would add an important element to the study and broaden the importance of the work.

We do appreciate the reviewer's comment. To demonstrate the utility of compound **6**, we applied it to study mitochondrial poly-ADP-ribosylation (PARylation). While the nucleus is known as the predominant subcellular location for protein PARylation, multiple studies indicated that PARylation may exist in mitochondria¹⁻⁶. And *in vitro* biochemical studies confirmed several mitochondrial proteins as substrates of PARP1⁷. Despite emerging but debating roles of PARylation in regulating mitochondrial DNA metabolism, there are limited approaches for studying mitochondrial PARylation. No activity-based probes are available to analyze mitochondrial PARylation. Given its high activity and selectivity for protein PARylation, compound **6** was attempted to label mitochondrial PARylation in cells permeabilized with 0.025% Triton X-100. Confocal microscopic analysis of cells treated with **6** showed that in addition to the predominant PARylation in nucleus, considerable PARylation signals were colocalized with mitofilin, a mitochondrial inner membrane protein (Figure 8A). The observed PARylation signals were suppressed by veliparib inhibitor. These results suggest the presence of PARylation in mitochondria.

We then isolated mitochondria from cells and confirmed the purity of the mitochondria fractions using a monoclonal anti-COX IV antibody for COX IV as mitochondrial protein loading controls, a monoclonal anti-GAPDH antibody for GAPDH as a cytoplasmic protein marker, and a polyclonal anti-Histone 2A.Z antibody for Histone 2A.Z as a nuclear protein marker. Consistent with confocal imaging results, immunoblot analysis of the isolated mitochondria fractions clearly revealed significant protein labeling in the presence of **6**. Treatment of cells with H₂O₂ resulted in increased protein labeling by **6** in the mitochondria fractions, which were suppressed by veliparib inhibitor (Figure 8B). To the best of our knowledge, this is the first time for visualization and labeling of PARylation in mitochondria using an activity-based probe. These results provide new and direct evidence for mitochondrial protein PARylation and demonstrate **6** as a valuable tool for studying cellular PARylation. The results and experimental methods for confocal microscopic and immunoblot analysis of mitochondrial PARylation by **6** are described on pages 15-16 and 26-27.

References:

1. Brunyanszki, A., Szczesny, B., Virag, L. & Szabo, C. Mitochondrial poly(ADP-ribose) polymerase: The Wizard of Oz at work. *Free Radic. Biol. Med.* **100**, 257-270 (2016).
2. Du, L. et al. Intra-mitochondrial poly(ADP-ribosylation) contributes to NAD⁺ depletion and cell death induced by oxidative stress. *J. Biol. Chem.* **278**, 18426-18433 (2003).
3. Lai, Y. et al. Identification of poly-ADP-ribosylated mitochondrial proteins after traumatic brain injury. *J. Neurochem.* **104**, 1700-1711 (2008).
4. Rossi, M.N. et al. Mitochondrial localization of PARP-1 requires interaction with mitofilin and is involved in the maintenance of mitochondrial DNA integrity. *J. Biol. Chem.* **284**, 31616-31624 (2009).
5. Szczesny, B., Brunyanszki, A., Olah, G., Mitra, S. & Szabo, C. Opposing roles of mitochondrial and nuclear PARP1 in the regulation of mitochondrial and nuclear DNA integrity: implications for the regulation of mitochondrial function. *Nucleic. Acids Res.* **42**, 13161-13173 (2014).
6. Brunyanszki, A., Olah, G., Coletta, C., Szczesny, B. & Szabo, C. Regulation of mitochondrial poly(ADP-Ribose) polymerase activation by the beta-adrenoceptor/cAMP/protein kinase A axis during oxidative stress. *Mol. Pharmacol.* **86**, 450-462 (2014).
7. Jiang, H., Kim, J.H., Frizzell, K.M., Kraus, W.L. & Lin, H. Clickable NAD analogues for labeling substrate proteins of poly(ADP-ribose) polymerases. *J. Am. Chem. Soc.* **132**, 9363-9372 (2010).

Reviewer 3

1. This study does not isolate and characterize the polymers that are produced by PARP using 3-N3-NAD as a substrate. Neither does it distinguish between the analogs relative rate of reaction for the initiation, the subsequent polymerization and the branching steps. It is therefore premature to conclude that high molecular branched polymers of similar structure to that formed using NAD as a substrate are the result when using the 3-N3-NAD. Clarification will require additional study, and in this communication the authors should simply and clearly point out that more work will be required to characterize the products of the reaction with the new substrate 3-N3-NAD.

We do appreciate the reviewer's comment. Further work is required to characterize the poly-ADP-ribose (PAR) polymers formed by compound **6**, including the individual steps of initiation, elongation, and branching for PARP1-catalyzed poly-ADP-ribosylation (PARylation) with **6** as the substrate. This is now described in the Discussion section in the second paragraph on page 17.

2. On page 5-6, yields of 68% are reported for the enzymatic conversion of NR6 to the NAD analog 6. It will be of interest to a reader to know if this refers to an isolated yield of 6 or to a yield calculated on the basis of HPLC analysis at the conclusion of the reaction. The authors should clarify the reporting of this result. It will also be of interest to know the maximum quantities of 6 that have been produced using this enzymatic synthesis.

We do appreciate the reviewer's comment. The yield of 68% for enzymatic synthesis of **6** from **NR6** is an isolated yield and calculated based on the weight of purified **6** by HPLC. At the end of the enzymatic reactions, **6** was purified by HPLC and fractions containing the desired product were combined, concentrated, and lyophilized to offer pure **6** as a colorless solid, which was used for the following experiments. We have clarified in the first paragraph on page 6 that the reported yields are isolated yields.

Upon confirming the facile production of **6** from **NR6** through the two-step enzymatic approach, the enzymatic reactions were scaled up under the same conditions and 12.2 mg of **6** was produced and purified for the later experiments, which is the maximum quantity of **6** we have produced through enzymatic synthesis. This information is now provided in the first paragraph on page 6.

3. Page 7 line 162-163. Lack of substrate activity for compound 5 could alternately be due to electronic properties of 5. The electron withdrawing 2-azide adjacent to the pyridinium-leaving group is predicted to decrease the rate of pyridine-ribose bond cleavage significantly. In a previous study [A.L.Hanlon et. al., JACS (1994) 116, 12087-12088], 2'-azido-NAD was shown to be resistant to both NAD glycohydrolase catalyzed and to chemical hydrolysis. The 2'-azide was hydrolyzed by the NAD glycohydrolase approximately 10,000-fold more slowly than was NAD. Compound 5 could therefore be a slowly cleaved or a non-cleavable substrate analog.

We do appreciate the reviewer's comment. The 2'-azido substitution is likely to increase the resistance of the adjacent *N*-glycosidic bond to chemical and enzymatic cleavage. Thus, the lack of substrate activity for compound **5** could be possibly caused by the blocked formation of branched PAR and/or the significantly decreased cleavage rate of the *N*-glycosidic bond resulting from electron-withdrawing 2'-azido group. This is now discussed in the first paragraph on page 17.

4. Page 14, lines 297-298. The experiment that the authors report demonstrates that **6** is not a substrate for SIRT2, not that there is an essential role for the nicotinamide ribotide 3'-OH in NAD analog **6**. An azide group is bigger and electronically different from an -OH, and the azide simply might not fit or might constrain that analog into a different confirmation. We can only conclude that compound **6** is not a substrate and that substitution of an -OH for an -N₃ is not tolerated by this enzyme.

We do appreciate the reviewer's comment and agree that we can only conclude that compound **6** is not a substrate of SIRT2 and azido substitution at 3'-OH of NR moiety is not tolerated by SIRT2. This has been revised in the third paragraph on page 17.

5. It is important that the authors define the purity of the compounds that are tested in this study. It is expected that tested compounds will be highly pure. Therefore, I would recommend the addition of a table or a paragraph to Methods that describes the purity of NAD analogs **1-6** and the method or methods that were used to assess the tested compounds purity. I expect (from the Supplementary Information) that the method will be analytical HPLC, but the purity of the compounds tested is never explicitly defined in the description of their synthesis.

We do appreciate the reviewer's comment. The purity of the reported compounds was determined by HPLC and all of them are highly pure with more than 97.9% HPLC purity. Each NAD⁺ analogue (500 μM) was analyzed separately by reverse-phase HPLC. The purity of **1-6** were then calculated based on the proportion of their corresponding integrated peak areas in total integrated peak areas. The HPLC chromatograms and determined purity of **1-6** are now shown in Figure S8 and Table S2, respectively, on page S20 in Supplementary Information. The method for the purity analysis of NAD⁺ analogues **1-6** is now described in the Methods section on pages 20-21. And we clarify that highly pure **1-6** were used for the study at the end of page 7.

6. The synthesis of the six NAD analogs described in this study begins with the construction of substituted ribose derivatives, the formation of a 1-bromo-ribose derivative, and the displacement of the 1-bromide with nicotinamide to form the nicotinamide-ribose derivative. The stereochemistry of the reaction product must be defined, and it's additionally important to insure that the NAD derivatives that are eventually synthesized contain the expected nicotinamide beta-riboside. Normally the beta-linkage is formed when the bromo-ribose has a 2-benzoyl substituent, but this is not the case for compound **5-7** and NR-**5** where a 2-azide is present. In this case and for **1-7** and **3-7** (containing 2-ether substituents) how was the stereochemistry of the products assigned, and how was it proved that the products are the beta-anomers as drawn?

We do appreciate the reviewer's comment. The stereochemistries of generated intermediates O-benzoyl protected NR**1-6** were determined as β-isomers on the basis of ¹H-¹H COSY experiments to confirm proton assignments and subsequent NOESY experiments (Figures S2 and S2-a-l). Since the configurations at C4 of compounds **1-7**, **2-7**, **3-7**, **4-6**, **5-7**, and **6-5** are the same as in the corresponding starting materials, the configurations at C1 of these compounds could be determined using NOESY spectroscopy. As shown in the Figures S2, S2-b, S2-d, S2-f, S2-h, S2-j and S2-l, the proton H¹ of each compound has no correlation with H³, H⁵, and H⁶, but correlates with the H⁴ proton. These results support the *cis* relationship between the H¹ and H⁴ and that the synthesized products contain desired nicotinamide beta-riboside. The COSY and NOESY spectra for the generated intermediates O-benzoyl protected NR**1-6** are shown on pages S8-14 in Supplementary Information.

Additionally, trimethylsilyl trifluoromethanesulfonate (TMSOTf) was used instead to introduce nicotinamide for synthesis of compounds **1-7**, **3-7**, and **5-7**. The relevant Schemes S1, S2, S4, and S6 and experimental details were accordingly revised in Supplementary Information. As shown in Figures S2, S2-b, S2-f, and S2-j, the configurations at C1 of **1-7**, **3-7**, and **5-7** were confirmed by NOESY experiments.

We feel that the manuscript is now suitable for publication in the *Nature Communications* and thank you for your time and effort on our behalf.

Reviewers' comments:

Reviewer #2 (Remarks to the Author):

I believe that the authors have responded to the concerns of the reviewers. I agree with the other reviewers that the strength of the paper is the new technology for preparing NAD⁺ analogs that appear to be quite efficiently used by PARP1.

The new data concerning mitochondrial poly(ADP-ribose) is a little underdeveloped in terms of explaining and interpreting the data. Is PARP1 the source of the poly(ADP-ribose) that is detected? and is there evidence for PARP1 in mitochondria? If not, what is the proposed source of the poly(ADP-ribose), and what does this mean for the specificity of compound 6? A somewhat related question: is PARP2, the closest relative of PARP1, capable of using compound 6? Also, the variations in active site loops that are proposed to underlie the differences in compound 6 utilization (Figure 5F), this really only represents one potential explanation. There very well could be more subtle differences in active site composition that could give rise to different efficiencies of using compound 6. Since the contribution of the active site loop is not directly tested, it is worth pointing out that other possibilities exist, in order to avoid giving the impression that the active site loop is the confirmed source of specificity compound 6 specificity.

If these questions could be answered through experiments or additional talking points in the text, I believe that the manuscript would be improved and better communicated.

Reviewer #3 (Remarks to the Author):

This manuscript successfully addresses a major issue in PARP research, and also significantly describes new methodology for the synthesis of NAD analogues. The revised manuscript successfully addresses all of the issues that have been raised during initial review, and I recommend that that paper now be published in Nature Communications.

James Slama, Ph.D.
Department of Medicinal and Biological Chemistry
University of Toledo

Reviewer 2

1. *The new data concerning mitochondrial poly(ADP-ribose) is a little underdeveloped in terms of explaining and interpreting the data. Is PARP1 the source of the poly(ADP-ribose) that is detected? and is there evidence for PARP1 in mitochondria? If not, what is the proposed source of the poly(ADP-ribose), and what does this mean for the specificity of compound 6?*

We do appreciate the reviewer's comment. Immunoblot analysis using an anti-human PARP1 antibody was performed for the isolated mitochondrial fractions. Consistent with several previous reports¹⁻³, a considerable amount of PARP1 was detected in mitochondrial fractions, suggesting that PARP1 could be an important contributor to protein PARylation in mitochondria. The immunoblot data are now included in Figure 8B on page 16 and these results are described and discussed in the first paragraph on page 15. The experimental method for immunoblot analysis of mitochondrial PARP1 is described on pages 26-27.

References:

1. Rossi, M.N. et al. Mitochondrial localization of PARP-1 requires interaction with mitofilin and is involved in the maintenance of mitochondrial DNA integrity. *The Journal of biological chemistry* **284**, 31616-31624 (2009).
2. Szczesny, B., Brunyanski, A., Olah, G., Mitra, S. & Szabo, C. Opposing roles of mitochondrial and nuclear PARP1 in the regulation of mitochondrial and nuclear DNA integrity: implications for the regulation of mitochondrial function. *Nucleic acids research* **42**, 13161-13173 (2014).
3. Brunyanski, A., Olah, G., Coletta, C., Szczesny, B. & Szabo, C. Regulation of mitochondrial poly(ADP-Ribose) polymerase activation by the beta-adrenoceptor/cAMP/protein kinase A axis during oxidative stress. *Molecular pharmacology* **86**, 450-462 (2014).

2. *A somewhat related question: is PARP2, the closest relative of PARP1, capable of using compound 6?*

We do appreciate the reviewer's comment. The substrate activities of compound **6**, **2**, 6-a-NAD⁺, and 2-a-NAD⁺ for PARP2 were examined using recombinant human PARP2 with an N-terminal GST tag purchased from Sigma-Aldrich through immunoblot analysis of PARP2-catalyzed auto-poly-ADP-ribosylation (PARylation). Similar to PARP1, human PARP2 modified by **6** showed significantly higher PARylation signals compared with 6-a-NAD⁺ and 2-a-NAD⁺. The immunoblot data for PARP2 activities are now shown in Figure 5D on page 12 and these results are described in the first paragraph on pages 11-12 and discussed in the second paragraph on page 17. The experimental method for analysis of substrate activities of **6**, **2**, 6-a-NAD⁺, and 2-a-NAD⁺ for PARP2 is described on pages 18 and 29-30.

3. *Also, the variations in active site loops that are proposed to underlie the differences in compound 6 utilization (Figure 5F), this really only represents one potential explanation. There very well could be more subtle differences in active site composition that could give rise to different efficiencies of using compound 6. Since the contribution of the active site loop is not directly tested, it is worth pointing out that other possibilities exist, in order to avoid giving the impression that the active site loop is the confirmed source of specificity compound 6 specificity.*

We do appreciate the reviewer's comment and agree that in addition to the highlighted active site loop, more subtle differences at the catalytic sites of those PARP enzymes could result in different activities with compound 6. These possibilities are now discussed in the second paragraph on page 17.

Reviewer 3

This manuscript successfully addresses a major issue in PARP research, and also significantly describes new methodology for the synthesis of NAD analogues. The revised manuscript successfully addresses all of the issues that have been raised during initial review, and I recommend that that paper now be published in Nature Communications.

We do appreciate the reviewer's comments.

REVIEWERS' COMMENTS:

Reviewer #2 (Remarks to the Author):

The authors have addressed the comments raised during the last review, regarding PARP2 activity, the mitochondrial PAR signal, and the structural suggestions for compound specificity.